# Quasi-Bayes meets Vines

**David Huk**
Department of Statistics
University of Warwick
David.Huk@warwick.ac.uk

**Yuanhe Zhang**
Department of Statistics
University of Warwick
Yuanhe.Zhang@warwick.ac.uk

**Mark Steel**
Department of Statistics
University of Warwick
m.steel@warwick.ac.uk

**Ritabrata Dutta**
Department of Statistics
University of Warwick
Ritabrata.Dutta@warwick.ac.uk

## Abstract

Recently developed quasi-Bayesian (QB) methods [29] proposed a stimulating change of paradigm in Bayesian computation by directly constructing the Bayesian predictive distribution through recursion, removing the need for expensive computations involved in sampling the Bayesian posterior distribution. This has proved to be data-efficient for univariate predictions, however, existing constructions for higher dimensional densities are only possible by relying on restrictive assumptions on the model's multivariate structure. Here, we propose a wholly different approach to extend Quasi-Bayesian prediction to high dimensions through the use of Sklar's theorem, by decomposing the predictive distribution into one-dimensional predictive marginals and a high-dimensional copula. We use the efficient recursive QB construction for the one-dimensional marginals and model the dependence using highly expressive vine copulas. Further, we tune hyperparameters using robust divergences (eg. energy score) and show that our proposed Quasi-Bayesian Vine (QB-Vine) is a fully non-parametric density estimator with *an analytical form* and convergence rate independent of the dimension of the data in some situations. Our experiments illustrate that the QB-Vine is appropriate for high dimensional distributions ($\sim$64), needs very few samples to train ($\sim$200) and outperforms state-of-the-art methods with analytical forms for density estimation and supervised tasks by a considerable margin.

## 1 Introduction

The estimation of joint densities is a cornerstone of machine learning as a looking glass into the underlying data-generating process of multivariate data. Methods that support explicit density evaluation are crucial in probabilistic modelling, with applications in variational methods [58, 85], Importance Sampling [2, 67], Sequential Monte Carlo [41], Markov Chain Monte Carlo (MCMC) [92, 49] and simulation-based inference [81, 62]. A prominent example are Normalising Flows (NF) [80, 24, 82], leveraging deep networks with invertible transformations for analytical expressions and sampling. Despite impressive performances, they require meticulous manual hyperparameter tuning and large amounts of data to train. Bayesian methods are another attractive approach for analytical density modelling where the central object of interest is the predictive density, with the Dirichlet Process Mixture Model (DPMM) [46] as the canonical nonparametric choice. Similar to kernel density estimation, the DPMM can be interpreted as an infinite mixture of densities. It is composed of a density called a kernel, with a fixed parametric form, and of a mixing density responsible for assigning those parameters. The specification of this kernel and mixing density is thus important

38th Conference on Neural Information Processing Systems (NeurIPS 2024).

as they regulate the expressivity of a DPMM. However, the computation of the DPMM's predictive density is laborious as it relies on expensive MCMC methods scaling poorly to high-dimensions[1].

Recently, [29] proposed an appealing shift in Bayesian methods through the efficient constructions of Quasi-Bayesian (QB) sequences of predictive densities using only recursion, thereby circumventing the need for MCMC in Bayesian inference, and inspiring an offspring of works utilising this methodology [28, 59, 35, 26, 99, 69, 101]. The seminal univariate Recursive Bayesian Predictive (R-BP) [43], its multivariate ($R_d$-BP) extension [29] and the AutoRegressive Bayesian Predictive (AR-BP) [35] are all QB models targeting the predictive mean of the DPMM, with an analytical recursive form driven by bivariate copula updates. Notably, the AR-BP demonstrated superior density estimation capabilities compared to a suite of competitors on varied supervised and unsupervised datasets, is orders of magnitude faster than standard Bayesian methods and is data-efficient, i.e. does not require large amounts of training data to be effective.

In multivariate QB predictives, to derive analytical expressions, it is necessary to enforce assumptions on the DPMM kernel structure. The kernel of the $R_d$-BP is set to be independent across dimensions, where this strong assumption is too constraining for more complex data. This is relaxed in the AR-BP through an autoregressive form of the kernel, where each kernel mean is a similarity function of previous dimensions modelled through a covariance function or a deep autoregressive network, depending on the complexity of the data. The former relies on a fixed form that is often too simplistic while the latter loses the appeal of a data-efficient predictive like in the $R_d$-BP. Here, we posit that these assumptions, required for obtaining existing recursions, lack the flexibility to model multivariate data accurately.

In this paper, we introduce the Quasi-Bayesian Vine (QB-Vine) as a more general approach to forming multivariate QB recursions, utilising a copula decomposition to circumvent restrictive assumptions on the DPMM kernel. The QB-Vine is obtained by applying Sklar's theorem [91] to the joint predictive, thereby dissecting it into univariate marginal predictive densities and a multivariate copula. Marginal predictives are modelled with the data-efficient univariate R-BP while the multivariate copula is modelled with a simplified vine copula - a highly flexible copula model suited for high dimensions. Compared to the sequential constructions of previous work, the QB-Vine is inherently parallelizable over dimensions instead of sequential. The main contributions of our work are as follows:

- The copula decomposition frees us from the need to assume specific kernel structures of the DPMM, but preserves the data efficiency of QB methods while making it more effective on high-dimensional data.

- Under certain assumptions on the true dependence structure within the data, we show that the QB-Vine attains a convergence rate that is independent of the dimension.

- The above decomposition of the joint density and use of the energy score to tune hyperparameters makes the QB-Vine amenable to efficient parallelisation with significant computational gains.

Our paper is structured as follows. In Section 2 we introduce Quasi-Bayesian prediction, recapitulating the R-BP construction. In Section 3 we formulate the Quasi-Bayesian Vine model. We provide a succinct survey of related work in Section 4 and compare related methods to the QB-Vine in Section 5 on a range of datasets for density estimation, regression, and classification, achieving state-of-the-art performance with our model. We conclude with a discussion in Section 6.

## 2 Quasi-Bayesian prediction

**Notation.** Let $\mathbf{p}(\mathbf{x})$ be a multivariate probability density function over $\mathcal{X} \subseteq \mathbb{R}^d$, from which we observe i.i.d. samples $\mathcal{D}_P = \{\mathbf{x}_k\}_{k=1}^n \sim \mathbf{p}(\mathbf{x})$. Similarly, let $p^1(x^1), \ldots, p^d(x^d)$ be the marginal densities of $\mathbf{p}(\mathbf{x})$, each over (a subset of) $\mathbb{R}$ with corresponding i.i.d. samples $\mathcal{D}_{P^i} = \{x^i{}_k\}_{k=1}^n \sim p^i(x^i)$ for $i = 1, \ldots, d$, and assumed to all be continuous. Further, let $\mathbf{P}$ and $P^1, \ldots, P^d$ be the respective cumulative distribution functions (cdfs) of the previously mentioned densities, in order of

---

[1]For reference, with $d$ the dimensionality of the parameter space, Random Walk Metropolis-Hastings scales like $\mathcal{O}(d^2)$ [87], Metropolis-adjusted Langevin algorithm like $\mathcal{O}(d^{5/4})$ [86], and Hamiltonian Monte Carlo like $\mathcal{O}(d^{4/3})$ [11]

appearance. Finally, when discussing predictive densities, we will use a subscript, plain for data (e.g. $x_n$) and in parentheses for functions (e.g. $p_{(n)}$) to indicate the predictive at step $n$, distinguishing them from the superscript kept for dimension, and use bold fonts exclusively for multivariate objects.

**Bayesian predictive densities as copula updates.** Consider the univariate Bayesian predictive density $p_{(n)}$ for a future observation $x \in \mathbb{R}$ given seen i.i.d. data $x_{1:n} \in \mathbb{R}^n$ with a likelihood $f$ of the data and a posterior $\pi_{(n)}$ for the model parameters $\theta$ after $n$ observations. By Bayes rule, we have:

$$p_{(n)}(x|x_{1:n}) = \frac{\int f(x|\theta) \cdot f(x_n|\theta) \cdot \pi_{(n-1)}(\theta|x_{1:n-1}) \, d\theta}{p_{(n-1)}(x_n|x_{1:n-1})}.$$

As discovered by [43, 29, 50], multiplying and dividing by the predictive from the previous step $p_{(n-1)}$, we arrive at

$$p_{(n)}(x|x_{1:n}) = p_{(n-1)}(x|x_{1:n-1}) \cdot \frac{\overbrace{\int f(x|\theta) \cdot f(x_n|\theta) \cdot \pi_{(n-1)}(\theta|x_{1:n-1}) \, d\theta}^{\text{Joint density for } x, \, x^n}}{\underbrace{p_{(n-1)}(x_n|x_{1:n-1})}_{\text{Marginal for } x_n} \cdot \underbrace{p_{(n-1)}(x|x_{1:n-1})}_{\text{Marginal for } x}} \tag{1}$$

$$= p_{(n-1)}(x|x_{1:n-1}) \cdot c_{(n)}\left(P_{(n-1)}(x), P_{(n-1)}(x_n)\right)$$

which by Sklar's theorem [91] identifies the involvement of copulas[2] in this recursive equation for the predictive densities. The second term on the right-hand side of (1) is seen to be a symmetric bivariate copula density function with the property that $\lim_{n\to\infty} c^{(n)}(x, x^n) = 1 \, a.s. \forall x$ as a consequence of the almost sure convergence of $p^{(n)}$ with $n$ [29]. It can be shown that every univariate Bayesian model can be written in this form [43], and so has a unique copula sequence characterising its predictive updates by de Finetti's theorem [20, 45]. The choice of the predictive sequence then corresponds to an implicit choice of likelihood and prior [10, 33].

**Marginal recursive Bayesian predictives.** Due to the integration over the posterior density often being intractable in practice, identifying the update copulas $c_{(n)}$ analytically is generally impossible. Therefore, [43] propose a nonparametric density estimator termed recursive Bayesian predictive (R-BP) as a Dirichlet Process Mixture Model (DPMM) inspired recursion emulating (1). They derive the correct Bayesian update under a DPMM for step 1 and use it for all future steps $m > 1$ (derivations shown in Appendix B.3). The update copula of the R-BP is a mixture between the independent and Gaussian copula, thereby deviating from the true (unknown) form of the Bayesian recursion copulas for a DPMM. For an initial choice of predictive density $p_{(0)}$ and distribution $P_{(0)}$, the obtained analytical expression for the R-BP recursion has the following predictive density

$$p_{(n)}(x) = p_{(n-1)}(x) \cdot \left[(1 - \alpha_n) + \alpha_n \cdot c_\rho(P_{(n-1)}(x), P_{(n-1)}(x_n))\right] \tag{2}$$

with $c_\rho \in [0, 1]$ being a bivariate Gaussian copula with covariance $\rho$. The corresponding cdf also admits an analytical expression as follows

$$P_{(n)}(x) = (1 - \alpha_n) \cdot P_{(n-1)}(x) + \alpha_n H_\rho(P_{(n-1)}(x), P_{(n-1)}(x_n)). \tag{3}$$

Here, for $\Phi$ the standard univariate Gaussian distribution,

$$H_\rho(u, v) = \Phi\left(\frac{\Phi^{-1}(u) - \rho \cdot \Phi^{-1}(v)}{\sqrt{1 - \rho^2}}\right)$$

is a conditional Gaussian copula distribution with covariance $\rho$ treated as a hyperparameter and where $\alpha_k = (2 - \frac{1}{k})\frac{1}{k+1}$ is a sequence of weights converging to 0 (See supplement E of [29] for a more detailed explanation of the weights). The computational cost is $\mathcal{O}(n^2)$ for initialising the recursion by computing $P_{(k)}(x_k)$ for $x_{1:n}$, and $\mathcal{O}(n)$ to evaluate the pdf or cdf at a point. This univariate R-BP model has been shown in [29] (Theorem 5) to converge in total variation to a limiting distribution $P_{(\infty)}$ with density $p_{(\infty)}$ termed the *martingale posterior*, defined in Appendix B, which is a quasi-Bayesian object describing current uncertainty [29, 31, 50, 8, 33]. [43] shows that the

---

[2]For an introduction to copulas, see Appendix A, [25, 40].

R-BP converges to the true density in the Kullback-Leibler distance. The recursive formulation of the R-BP with an analytical form ensures fast updates of predictives whilst the use of copulas bypasses the need to evaluate a normalising constant (as is the case in Newton's algorithm, a similar recursion for a DPMM's predictive [75, 74]). Consequently, the R-BP is free from the reliance on MCMC to approximate a posterior density, making it much faster than regular DPMMs. While this formulation does not correspond to a Bayesian model, as argued by [7, 10, 29, 50], if the recursive updates are conditionally identically distributed (as is the case for the recursion of (3)), they still exhibit desirable Bayesian characteristics such as coherence, regularization, and asymptotic exchangeability, motivating the Quasi-Bayesian name as used in [30].

**Multivariate recursive Bayesian predictives.** An extension to the multivariate case was studied by [29] and refined by [35] for data with more complex dependence structures. The multivariate DPMM is formulated as

$$f(\mathbf{x} \mid G) = \int_\Theta K(\mathbf{x} \mid \boldsymbol{\theta}) dG(\boldsymbol{\theta}), \text{ with } G \sim \mathrm{DP}\,(c, G_0)$$

where $K$ is a kernel for the observables $\mathbf{x} \in \mathbb{R}^d$ parameterised by $\boldsymbol{\theta}$, similarly to the kernel in kernel density estimation, and $G$ is called the mixing distribution, upon which a Dirichlet process prior is placed with base measure $G_0$ and concentration parameter $c > 0$. To address this shortcoming, [29] provide a copula-based recursion obtained by assuming $K(\mathbf{x}|\boldsymbol{\theta}) = \prod_{i=1}^d \mathcal{N}(x^i|\theta^i, 1)$ and $G_0(\boldsymbol{\theta}) = \prod_{i=1}^d \mathcal{N}(\theta^i|0, \tau^{-1})$, $\tau > 0$, meaning both the kernel and base measure are assumed independent across dimensions, lacking the expressivity required to capture dependencies in the data. In [35], the form of the kernel is relaxed to be autoregressive with $K(\mathbf{x}|\boldsymbol{\theta}) = \prod_{i=1}^d \mathcal{N}(x^i|\theta^i(\mathbf{x}^{1:i-1}), 1)$ where the kernel mean $\theta^i : \mathbb{R}^{i-1} \mapsto \mathbb{R}$ is dependent on previous dimensions, and the base measure of [31] is a product of Gaussian Process priors $G_0(\boldsymbol{\theta}) = \prod_{i=1}^d GP(\theta^i|0, \tau^{-1}k)$ for $k : \mathbb{R}^{i-1} \times \mathbb{R}^{i-1} \mapsto \mathbb{R}$ a covariance function.

**Vine copulas as effective models for high dimensions.** Vine copulas are a class of copulas that provide a divide-and-conquer approach to high-dimensional modelling by decomposing the joint copula density into $\frac{d(d-1)}{2}$ bivariate copula terms. They are considered among the best current copula models for density estimation. The main ingredient of a vine copula decomposition is the following identity as a consequence of Sklar's theorem [91]:

$$p^{a|b}(x^a|x^b) = c^{a,b}(P^a(x^a), P^b(x^b)) \cdot p^a(x^a) \tag{4}$$

where $a, b$ are subsets of dimensions from $\{1, \dots, d\}$. Vine copulas rely on a conditional factorisation $\mathbf{p}(x^1, \dots, x^d) = \prod_{i=1}^d p^{i|1:i-1}(x^i|x^{1:i-1})$ to which they repeatedly apply (4), thereby splitting the joint density into the $d$ marginal densities and $\frac{d(d-1)}{2}$ bivariate copulas called pair copulas. The pair copulas for each $i \neq j \in \{1, \dots, d\}$, take as input pairs of conditional distributions $(P^{i|\mathcal{S}^{ij}}(x^i|\mathcal{S}^{ij}), P^{j|\mathcal{S}^{ij}}(x^j|\mathcal{S}^{ij}))$ where $\mathcal{S}^{ij} \subseteq \{1, \dots, d\} \setminus \{i, j\} \cup \emptyset$ is decided by the choice of the vine. A vine copula model thus has the form

$$\mathbf{c}(P^1(x^1), \dots, P^d(x^d)) = \prod_{i \neq j}^{d(d-1)/2} c^{ij}(P^{i|\mathcal{S}^{ij}}(x^i|\mathcal{S}^{ij}), P^{j|\mathcal{S}^{ij}}(x^j|\mathcal{S}^{ij})|\mathcal{S}^{ij}). \tag{5}$$

We notice, that these pair copulas start as unconditional bivariate copulas and later capture higher orders of multivariate dependence by conditioning on the set $\mathcal{S}$ itself. This decomposition is valid but only unique up to the permutation of indexes. We provide an example of a three-dimensional vine copula decomposition and an overview in Appendix A.3, referring the reader to [17, 18] for an introduction. In practice, we use a *simplified vine copula* model [70, 71] which removes the conditional dependence of each of the copula $c^{ij}$ on $\mathcal{S}^{ij}$. This is an approximation which reduces the complexity of the model for dependency structure but provides significant computational gains by reducing the size of the model space. An example simplified vine model in $d = 3$ is shown below:

$$\mathbf{c}(P^1(x^1), P^2(x^2), P^3(x^3)) = c^{12}(P^1(x^1), P^2(x^2)) \cdot c^{13}(P^1(x^1), P^3(x^3)) \cdot c^{2,3|1}(P^2(x^2|x^1), P^3(x^3|x^1)).$$

The number of pair copulas grows quadratically with the dimension, and the number of possible decompositions is exponential with the dimension, leading to greedy algorithms being used for model selection, see [21]. The flexibility and efficacy of the vine has been studied in the literature, we refer to [17, 18, 71–73, 93] among others. In particular, when the simplified vine assumption is true, [70] provides a dimension-independent convergence rate making simplified vine copulas greatly appealing for high-dimensional models.

# 3 Quasi-Bayesian Vine prediction

We propose the Quasi-Bayesian Vine (QB-Vine) for efficiently modelling a high-dimensional predictive density $\mathbf{p}_{(n)}$ (and distribution $\mathbf{P}_{(n)}$). The efficiency is achieved by adapting Sklar's theorem ([91], see Appendix A) to split the joint predictive into predictive marginals for each dimension and a high-dimensional copula:

**Theorem 3.1.** *Let $\mathbf{P}_{(n)}$ be an $d$-dimensional predictive distribution function with continuous marginal predictive distributions $P_{(n)}^1, P_{(n)}^2, \ldots, P_{(n)}^d$. Then there exists a copula distribution $\mathbf{C}_{(\mathbf{n})}$ such that for all $\mathbf{x} = (x^1, x^2, \ldots, x^d) \in \mathbb{R}^d$:*

$$\mathbf{P}_{(n)}(x^1, \ldots, x^d) = \mathbf{C}_{(n)}(P_{(n)}^1(x^1), \ldots, P_{(n)}^d(x^d))$$

*And if a probability density function is available:*

$$\mathbf{p}_{(n)}(x^1, \ldots, x^d) = p_{(n)}^1(x^1) \cdot \ldots \cdot p_{(n)}^d(x^d) \cdot \mathbf{c}_{(n)}(P_{(n)}^1(x^1), \ldots, P_{(n)}^d(x^d)) \tag{6}$$

*where $p_{(n)}^1(x^1), \ldots, p_{(n)}^d(x^d)$ are the marginal predictive probability density functions, and $\mathbf{c}_{(n)} : [0,1]^d \to \mathbb{R}$ is the copula probability density function.*

By applying the decomposition in (6) to two consecutive predictive densities $\mathbf{p}_{(m-1)}$ and $\mathbf{p}_{(m)}$, we obtain a recursive update for joint predictive densities with two parts, of the form:

$$\frac{\mathbf{p}_{(m)}(\mathbf{x})}{\mathbf{p}_{(m-1)}(\mathbf{x})} = \underbrace{\prod_{i=1}^d \left\{ \frac{p_{(m)}^i(x^i)}{p_{(m-1)}^i(x^i)} \right\}}_{\text{Independent recursions}} \cdot \underbrace{\frac{\mathbf{c}_{(m)}\left(P_{(m)}^1(x^1), \ldots, P_{(m)}^d(x^d)\right)}{\mathbf{c}_{(m-1)}\left(P_{(m-1)}^1(x^1), \ldots, P_{(m-1)}^d(x^d)\right)}}_{\text{Implicit recursion on copulas}} .$$

This decomposition fruitfully isolates updates to marginal predictive densities from updates to their dependence structure, allowing us to model each recursion separately; the marginal predictives follow a univariate recursion *a la* (1) while the copulas are free to follow another recursive form. Particularly:

- As we are only interested in the joint predictive $\mathbf{p}_{(n)}$, once marginal predictives are obtained, we only need to fit a single copula $\mathbf{c}_{(n)}$ at step $n$ to recover the joint predictive through (6).
- Unlike [29, 35] where the recursion is done sequentially across dimensions, the QB-Vine can recurse marginal predictives in parallel by dimension.
- The model's dependence is not constrained by assumptions on the DPMM's form, instead, the QB-Vine is free to fit any copula that best matches the dependence of the data.

**Marginal predictive density estimation with the R-BP.** We model marginal predictive densities $p_{(n)}^i(x_i)$ and distributions $P_{(n)}^i(x_i), \forall i = 1, \ldots, d$ independently between dimensions. We use the univariate R-BP approach described in Section 2 to recursively obtain the analytical expression for both. For each dimension separately, starting with an initial density $p_{(0)}^i$ and distribution $P_{(0)}^i$, we follow the updates (2) for the density and (3) for the distribution.

**Simplified vine copulas for high-dimensional dependence.** After estimating marginal predictives, we model the joint density of $\left(u^1 := P^1_{(n)}(x^1), \ldots, u^d := P^d_{(n)}(x^d)\right)$ with a multivariate copula. We consider a highly flexible simplified vine copula, found in Equation (5), which decomposes the joint copula density $c(u^1, \ldots, u^d)$ into $\frac{d(d-1)}{2}$ bivariate copulas $c^{ij}$ of the cdfs from dimensions $i$ and $j$ (possibly conditioned on additional dimensions) to capture the dependence structure of $\mathbf{x}$. For the bivariate pair-copulas $c^{ij}$, we use a nonparametric Kernel Density Estimator (KDE)[3]. Thus, each $c^{ij}$ becomes a two-dimensional KDE copula with the following expression:

$$c^{ij}(u, v) = \frac{\sum_{k=1}^n \phi\left(\Phi^{-1}(u) - \Phi^{-1}(u_k^i); 0, b\right) \cdot \phi\left(\Phi^{-1}(v) - \Phi^{-1}(v_k^j); 0, b\right)}{\phi\left(\Phi^{-1}(u); 0, b\right) \cdot \phi\left(\Phi^{-1}(v); 0, b\right)}$$

where $\phi(.; 0, b)$ is the pdf of a normal with mean 0 and variance $b$, $\Phi^{-1}$ is the inverse standard normal cdf. Samples $\{(u_k^i, v_k^j)\}_{k=1}^n$ are easily obtained by iteratively fitting KDE pair copulas on observed samples $\{(u_k^1, \ldots, u_k^d)\}_{k=1}^n$ [17].

---

[3]For further detail on KDE-based vine copulas, see Appendices A.2, A.3 and [70, 71].

**Algorithm 1** Joint, marginal, and copula density estimation with the Quasi-Bayesian Vine

**Input:** $\texttt{train}(M, N', d), \texttt{test}(N, d), [B_1, \ldots, B_l], V, J.$

**Marginals:**
1: **for** $i = 1$ to $d$ **do**      // Dimension-wise training and evaluation, **parallelised**
2:      **repeat**
3:          **for** $m = 1$ to $M$ **do**      // average over permutations, **parallelised**
4:              **for** $n = 1$ to $N'$ **do**
5:                  ● Compute $v_n^{i,m} := P_{(n)}^{i,m}(\texttt{train}_n^{i,m}|v_{1:(n-1)}^{i,m})$ recursively using (3)
6:              **end for**
7:          **end for**
8:          ● Set $P_{(N'+1)}^i(\cdot) = \Sigma_{m=1}^M P_{(N'+1)}^{i,m}(\cdot|v_{1:N'}^{i,m})/M$
9:          ● Compute the energy score $\mathcal{S}_E(x_{1:J}^i\|\texttt{train}^i)$ for $x_{1:J}^i \sim P_{(N'+1)}^i(\cdot)$, using (9)
10:          ● Update $\rho^i$ based on $\mathcal{S}_E$
11:      **until** convergence of $\rho^i$
12:      **for** $n = 1$ to $N'$ **do**
13:          ● Evaluate $P_{(N'+1)}^i(\{\texttt{train}^i, \texttt{test}^i\})$ recursively using (3)
14:          ● Evaluate $p_{(N'+1)}^i(\{\texttt{train}^i, \texttt{test}^i\})$ recursively using (2)
15:      **end for**
16: **end for**

**Copula Fitting:**
1: **for** $b = B_1$ to $B_l$ **do**      // Optimisation, **parallelised**
2:      **for** $v = 1$ to $N' \bmod(V)$ **do**      // cross-validation, **parallelised**
3:          ● Fit vine $\left(P_{(N'+1)}^{1:d}(\texttt{train}_{1:V}^{1:d})|b\right)$
4:          ● Compute the energy score $\mathcal{S}_E(b) = S_E\left(u_{1:J}^{1:d}\|P_{(N')}^{1:d}(\texttt{train}_{(V+1):N'}^{1:d})\right)$ for $u_{1:J}^{1:d} \sim \text{vine}(\cdot|b)$, using (8)
5:      **end for**
6: **end for**
7: ● Select $b^* = \arg\min_b \mathcal{S}_E(b)$ for $b \in [B_1, \ldots, B_l]$
8: ● Evaluate $\mathbf{v}_{(N'+1)}(\texttt{test}) = \text{vine}(P_{(N'+1)}^{1:d}(\texttt{test})|b^*)$ using (5)
9: ● Evaluate $\mathbf{p}_{(N'+1)}(\texttt{test}) = \mathbf{v}_{(N'+1)}(\texttt{test}) \cdot \prod_{i=1}^d \{p_{(N'+1)}^i(\texttt{test}^i)\}$

**Return:** $\mathbf{p}_{(N'+1)}(\texttt{test}), p_{(N'+1)}^{1:d}(\texttt{test}), P_{(N'+1)}^{1:d}(\texttt{test}), \mathbf{v}_{(N'+1)}(\texttt{test}), \rho^{1:d}, b^*$

**Choice of Hyperparameters.** We begin by choosing a Cauchy distribution as the initial predictive distribution $p_{(0)}^i$ and the corresponding density $p_{(0)}^i \ \forall i = 1, \ldots, d$, with details provided in Appendix E. The hyperparameter $\rho$ for the R-BP recursion is assumed different for each marginal predictive (*i.e.* $\rho^d$). Unlike previous R-BP works [43, 29, 35], the QB-Vine minimize the energy score to select $\rho^d$ rather than the log-score, both of which are strictly proper scoring rules (see Appendix C and [78]) and define statistical divergences. This choice was motivated by the robustness properties of the energy score [77]. The energy score is computed between observations and $J = 100$ predictive samples from our marginal models conditional on previously observed data. As the R-BP is sensitive to the ordering of the data, we follow [29, 35] by averaging the resulting R-BP marginal over $M = 10$ permutations of the data (see [97, 22] for a discussion regarding the need of averaging over permutations).

We assume a same variance parameter $b$ for all the KDE pair copula estimators in the simplified vine and select it using 10-fold cross-validation, in a data-dependent manner by minimizing the energy score between observations and J=100 copula samples. The assumption of a common bandwidth $b$ is motivated by mapping all pair copulas to a Gaussian space, which results in a common distance used on the latent pair copula spaces. Another hyperparameter is the specific vine decomposition (the grouping of dimensions in (5), see Appendix A.3) for which we use a regular vine structure [79], selecting the best pair-copula decomposition with a modified Bayesian Information Criterion suited for vines [72].

We include an algorithmic description for estimating the marginal density as well as the copula with the QB-Vine in Algorithm 1, where $M$ is the number of permutations, $N'$ and $N$ the train and test sizes, $d$ the dimension of the data, $[B_1, \ldots, B_l]$ the copula variances considered for $b$, $V$ the cross-validation size, and $J$ the sample size for computing the energy score.

**Computational benefits of the QB-Vine.** Optimising the energy score instead of the log-score, together with the copula decomposition, provides us with some significant computational gains.

Firstly, as the energy score is a sample-based metric, we compute it through efficient univariate inverse probability sampling of $P^i_{(n)}$. Thus, we only recurse cdfs $P^i_{(n)}$ during training, thereby halving the time to tune hyper-parameters compared to using the log-score which requires densities $p^i_{(n)}$ on top of cdfs $P^i_{(n)}$ in (2). Secondly, Sklar's theorem implies the independence of marginal densities in the QB-Vine, allowing us to model them in parallel, reducing the cost to that of a single R-BP, *i.e.* constant with $d$. Finally, the vine hyperparameter $b$ is selected with a grid search using cross-validation, and parallelised across the grid and across cross-validation folds, each having the cost of a single vine.

**Properties of the Quasi-Bayesian Vine.**    To quantify the approximation of the QB-Vine, we provide the following stochastic boundedness [12] result for univariate R-BP distributions with respect to the limiting *martingale posterior* $P_{(\infty)}$ (see [29] and Appendix B). We note $P_{(\infty)}$ is the univariate martingale posterior of the R-BP. The multivariate martingale posterior of the QB-Vine is not used in our results, but we discuss its properties, including a martingale condition, in Appendix B.2.

**Lemma 3.2.** *(R-BP predictive distribution convergence) The error of the distribution function $P_{(n)}(x)$ in (3) is stochastically bounded with*

$$\sup_{x \in \mathcal{X}} \left| P_{(\infty)}(x) - P_{(n)}(x) \right| = \mathcal{O}_p \left( n^{-1/2} \right).$$

Appendix D.1 gives a proof. In comparison to univariate KDE with a mean-square optimal bandwidth $b_n = \mathcal{O}(n^{-1/5})$, which converges at a rate $\mathcal{O}_{a.s.}(n^{-2/5}\sqrt{\ln(n)})$, the marginal R-BP has a better rate with sample size. In what follows, we assume that the true copula is a simplified vine of which we know the decomposition (a standard assumption in the vine copula literature [70, 18, 93]). We strengthen marginal guarantees with the theory on vine copulas to obtain the following convergence result for the estimate of the copula density. In the statement of the theorem, we consider marginal distributions $\{P^i_{(\infty)}\}^d_{i=1}$ and $\{P^i_{(n)}\}^d_{i=1}$ are implicitly applied to $\mathbf{x}$ for respective copulas.

**Theorem 3.3.** *(Convergence of Quasi-Bayesian Vine) Let $\mathbf{c}_{(\infty)}(\mathbf{u})$ be the copula of $\{P^i_{(\infty)}(x^i)\}^d_{i=1}$ and let $\mathbf{c}_{(n)}(\mathbf{u})$ be the copula of $\{P^i_{(n)}(x^i)\}^d_{i=1}$. Assuming that both copulas are simplified vine copulas with the same permutation of indexes in the decomposition of (5), the estimation error is stochastically bounded $\forall\, \mathbf{x} \in \mathbb{R}^d$ with*

$$|\mathbf{c}_{(\infty)}(\mathbf{x}) - \mathbf{c}_{(n)}(\mathbf{x})| = \mathcal{O}_p(n^{-r})$$

*where $n^{-r}$ is the convergence rate of the KDE pair-copula.*

We provide a proof in Appendix D.2. For a bivariate KDE pair-copula estimator with optimal bandwidth $b_n = \mathcal{O}\left(n^{-1/6}\right)$, we obtain $n^{-r} = n^{-1/3}$ [70]. From [95], we note the optimal convergence rate of a nonparametric estimator is $n^{-q/(2q+q)}$ where $q$ is the number of times the estimator is differentiable. Therefore, as $d$ increases, we expect large benefits from using a vine copula decomposition for the QB-Vine. When the simplifying assumption does not hold, the simplified vine copula converges to a partial vine approximation of the true copula, as defined in [93]. Together, these two results guarantee accurate samples from the QB-Vine by inverse probability sampling arguments. By Theorem 3.3, the copula $\mathbf{c}_{(n)}$ ensures samples $\mathbf{u} = (u^1, \dots, u^d)$ on the $[0,1]^d$ hypercube have a dependence structure representative of the data. Then, marginal distributions $\{p^i_{(n)}\}^d_{i=1}$ recover dependent samples $\mathbf{x} \in \mathbb{R}^d$ by evaluating the inverse of the distribution at $u^i$ dimension-wise.

**Adapting the QB-Vine for Regression/classification tasks.**    Our framework can accommodate regression and classification tasks in addition to density estimation by rewriting the conditional density as following:

$$p(y|\mathbf{x}) = \frac{p(y, \mathbf{x})}{p(\mathbf{x})} = \frac{p_y(y) \cdot \prod_{i=1}^d \left\{ p^i(x^i) \right\} \cdot \mathbf{c}(y, x^1, \dots, x^d)}{\prod_{i=1}^d \left\{ p^i(x^i) \right\} \cdot \mathbf{c}(x^1, \dots, x^d)} = \frac{\mathbf{c}(y, x^1, \dots, x^d) \cdot p_y(y)}{\mathbf{c}(x^1, \dots, x^d)}. \quad (7)$$

The estimation of (7) is comprised of estimating the $d+1$ marginals for $y, x^1, \dots, x^d$ and the two copulas $\mathbf{c}(y, \mathbf{x})$ and $\mathbf{c}(\mathbf{x})$. We specify separate $\rho$ across marginal densities as well as separate

bandwidths $b$ for the two copulas, to be estimated as in Algorithm 1. We note that for a copula decomposition to be unique, we require that the marginals involved be continuous. This assumption is violated in the classification for binary outcomes $y$. As such, we make use of an approximation that transforms $y$ to a continuous scale by setting negative examples to $-10$, and positive examples to $10$ and adds standard Gaussian noise to all examples, breaking ties on a distribution scale (a common approach taken in similar contexts [55, 61, 44]). The rationale behind this approximation is that by setting the two classes far apart on a marginal scale, we ensure no overlaps occur, thereby maintaining a clear cut-off between the classes on the distribution scale. Indeed, the separating boundary between the classes on a distribution scale will be the percentile $q = \frac{T_0}{T_1+T_0}$ where $T_0$ and $T_1$ are the numbers of negative and positive samples, respectively, in the training set. Consequently, we create different clusters in the copula $[0, 1]^d$ hypercube according to the separation on the distributional scale, facilitating the identification of patterns in the data. We note that other approaches exist in the literature [14, 15, 18] for classification with copulas which our framework can be extended to.

## 4 Related Work

Our method shares similarities with existing work on QB predictive density estimation with analytical forms. The pivotal works of [75, 74] and the ensuing Predictive Recursion (PR) [37, 65, 66, 97, 63, 36, 64] propose a recursive solution to the same problem but are restrained to low dimensional settings due to the numerical integration of a normalising constant over a space scaling with $d$. A sequential importance sampling strategy for PR is proposed in [23] termed as PRticle Filter. The R-BP of [43] and the multivariate extensions in [29, 35] also have a recursive form driven by bivariate copula updates. In the multivariate case, imposing assumptions on the kernel structure leads to a conditional factorisation of the joint predictive which recovers bivariate copula updates. In [35], an autoregressive Bayesian predictive (AR-BP) is used, where the dependence is captured by dimension-wise similarity functions modelled with kernels or deep autoregressive networks. The former relies on assumptions that might be too simplistic to capture complex data while the latter loses the appeal of a data-efficient predictive like in the R-BP. The Quasi-Bayesian Vine retains the advantages of the bivariate copula-based recursion for marginal predictives and circumvents the need for assumptions on the DPMM kernel. We achieve this via approximating the high-dimensional dependency through a simplified vine copula which is highly flexible and does not use a deep network to preserve data-efficiency, all the while maintaining an analytical expression. A relevant benchmark are the NFs of [80, 24] with analytical densities with a state-of-the-art performance across data types and tasks.

## 5 Experiments

In this section, we compare our QB-Vine model against competing methods supporting density evaluation with a closed-form expression. Further details on the experiments are included in Appendix E. Code is included at `https://github.com/Huk-David/QB-Vine`.

**Density estimation.** We evaluate the QB-Vine on density estimation benchmark UCI datasets [4] with small sample sizes ranging from $89$ to $506$ and dimensionality varying from $12$ to $30$, adding results for the QB-Vine and PRticle Filter to the experiments of [35]. We report the log predictive score LPS$= \frac{1}{n_{test}} \sum_{k=1}^{n_{test}} -\ln\left(\mathbf{p}_{(n_{train})}(\mathbf{x}_k)\right)$ on a held-out test dataset of size $n_{test}$ comprised of half the samples with the other half used for training, averaging results over five runs with random partitions each time. We compare the QB-Vine against the following models: Kernel Density Estimation [83], DPMM [84] with a diagonal (Diag) and full (Full) covariance matrix for each mixture component, MAF [80], RQ-NSF [24] as well as the closely related PRticle Filter [23], R-BP [29] and AR-BP [35]. For the last two Bayesian predictive models, we add a subscript $d$ to indicate that the $\rho$ hyperparameter possibly differs across dimensions, and the $net$ suffix indicates a network-based selection of $\rho$ for dimensions. We observe in Table 1 that our QB-Vine method comfortably outperforms all competitors as the dimension increases, while getting close to the performance of the best alternative Bayesian predictive models for the lower dimensional WINE dataset. Our method's relative performance increases with the dimension of the data, particularly achieving a much smaller LPS for IONO - the dataset with the largest dimensions and a relatively small sample size. We accredit this performance to the copula decomposition as that is our main distinguishing factor from the other Bayesian predictive models.

Table 1: Average log predictive score (lower is better) with error bars corresponding to two standard deviations over five runs for density estimation on datasets analysed by [35]. We note that as dimension increases, the QB-Vine outperforms all benchmarks.

| $n/d$ | WINE 89/12 | BREAST 97/14 | PARKIN 97/16 | IONO 175/30 | BOSTON 506/13 |
|---|---|---|---|---|---|
| KDE | $13.69_{\pm 0.00}$ | $10.45_{\pm 0.24}$ | $12.83_{\pm 0.27}$ | $32.06_{\pm 0.00}$ | $8.34_{\pm 0.00}$ |
| DPMM (Diag) | $17.46_{\pm 0.60}$ | $16.26_{\pm 0.71}$ | $22.28_{\pm 0.66}$ | $35.30_{\pm 1.28}$ | $7.64_{\pm 0.09}$ |
| DPMM (Full) | $32.88_{\pm 0.82}$ | $26.67_{\pm 1.32}$ | $39.95_{\pm 1.56}$ | $86.18_{\pm 10.22}$ | $9.45_{\pm 0.43}$ |
| MAF | $39.60_{\pm 1.41}$ | $10.13_{\pm 0.40}$ | $11.76_{\pm 0.45}$ | $140.09_{\pm 4.03}$ | $56.01_{\pm 27.74}$ |
| RQ-NSF | $38.34_{\pm 0.63}$ | $26.41_{\pm 0.57}$ | $31.26_{\pm 0.31}$ | $54.49_{\pm 0.65}$ | $-2.20_{\pm 0.11}$ |
| PRticle Filter | $23.89,_{\pm 0.93}$ | $25.98_{\pm 1.06}$ | $34.79_{\pm 3.95}$ | $79.22_{\pm 9.87}$ | $27.18_{\pm 3.12}$ |
| R-BP | $13.57_{\pm 0.04}$ | $7.45_{\pm 0.02}$ | $9.15_{\pm 0.04}$ | $21.15_{\pm 0.04}$ | $4.56_{\pm 0.04}$ |
| $R_d$-BP | $13.32_{\pm 0.01}$ | $6.12_{\pm 0.05}$ | $7.52_{\pm 0.05}$ | $19.82_{\pm 0.08}$ | $-13.50_{\pm 0.59}$ |
| AR-BP | $13.45_{\pm 0.05}$ | $6.18_{\pm 0.05}$ | $8.29_{\pm 0.11}$ | $17.16_{\pm 0.25}$ | $-0.45_{\pm 0.77}$ |
| $AR_d$-BP | $\mathbf{13.22_{\pm 0.04}}$ | $6.11_{\pm 0.04}$ | $7.21_{\pm 0.12}$ | $16.48_{\pm 0.26}$ | $-14.75_{\pm 0.89}$ |
| ARnet-BP | $14.41_{\pm 0.11}$ | $6.87_{\pm 0.23}$ | $8.29_{\pm 0.17}$ | $15.32_{\pm 0.35}$ | $-5.71_{\pm 0.62}$ |
| QB-Vine | $13.76_{\pm 0.13}$ | $\mathbf{4.67_{\pm 0.31}}$ | $\mathbf{4.93_{\pm 0.20}}$ | $\mathbf{-16.08_{\pm 2.12}}$ | $\mathbf{-31.04_{\pm 1.02}}$ |

**High-dimensional image dataset.** We further evaluate the QB-Vine on the digits dataset ($n =$1797, $d$=64) as a high-dimensional example with a relatively low sample size. The high contrast between $n$ and $d$ makes the problem suited for assessing the data efficiency and convergence of the QB-Vine. We compare with the two NF models as their high model capacity is a good fit for image data, as well as all the Bayesian predictive methods of Section 5, from the study of [35]. We report the average LPS in bits per dimension (bpd) with standard errors over fifteen runs with random partitions, using half the sample size to train models and the other half to evaluate the LPS. Additionally, we report the average LPS of the QB-Vine, obtained in the same way except for the training set size being reduced (to 30, 50, 100, 200, 300, 400, 500). Figure 1 depicts the QB-Vine's performance for different-sized training sets. When trained on

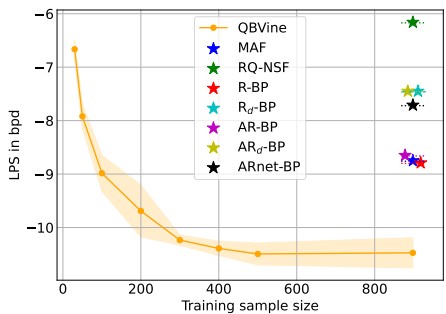

Figure 1: Density estimation on the Digits data ($n = 1797, d = 64$) with reduced training sizes for the QB-Vine against other models fitted on the full training set. The QB-Vine achieves competitive performance for training sizes as little as $n = 50$ and outperforms all competitors once $n > 200$.

the full training set, the QB-Vine outperforms all competitors by a considerable margin. Furthermore, our method is competitive with as little as 50 training samples and outperforms all benchmarks past a training size of 200, demonstrating its data-efficiency and convergence speed. A complete numerical table is reported in Appendix E.

**Regression and classification.** We further demonstrate our method's effectiveness on supervised learning tasks, with three datasets for regression and two datasets for classification, adding to the study of [35]. For classification, we transform the binary values to continuous ones to preserve copula assumptions, as detailed in Section 3. We report the conditional LPS $= \frac{1}{n_{test}} \sum_{k=1}^{n_{test}} - \ln \big( \mathbf{p}_{(n_{train})}(y_k|\mathbf{x}_k) \big)$ over a test set of size $n_{test}$ made up of half the samples with the conditional estimator trained on the other half of the data. We compare our model against a Gaussian Process [100], a linear Bayesian model (Linear) [68], a one-hidden-layer multilayer perceptron (MLP), as well the R-BP and AR-BP variants for supervised tasks [29, 35]. The QB-Vine outperforms competing methods on all datasets except CONCR. We believe the lower performance on CONCR is due to the high number of samples relative to the dimension, preventing our approach from fully exploiting the vine copula decomposition. Once again, the performance of the QB-Vine more clearly exceeds that of competitors as dimensions increase. The QB-Vine has higher standard errors than other methods (except MLP), which we posit is the consequence of our conditional estimator in Section 3 being defined as a ratio, inflating the variation in the LPS. However, we highlight that an overly precise inference is more misleading/dangerous than an overly uncertain one.

Table 2: Average LPS (lower is better) with error bars corresponding to two standard deviations over five runs for supervised tasks analysed by [35]. The QB-Vine performs favourably against benchmarks, with relative performance improving as samples per dimension decrease.

| | Regression | | | Classification | |
|---|---|---|---|---|---|
| | BOSTON | CONCR | DIAB | IONO | PARKIN |
| $n/d$ | 506/13 | 1,030/8 | 442/10 | 351/33 | 195/22 |
| Linear | $0.87_{\pm 0.03}$ | $0.99_{\pm 0.01}$ | $1.07_{\pm 0.01}$ | $0.33_{\pm 0.01}$ | $0.38_{\pm 0.01}$ |
| GP | $0.42_{\pm 0.08}$ | $0.36_{\pm 0.02}$ | $1.06_{\pm 0.02}$ | $0.30_{\pm 0.02}$ | $0.42_{\pm 0.02}$ |
| MLP | $1.42_{\pm 1.01}$ | $2.01_{\pm 0.98}$ | $3.32_{\pm 4.05}$ | $0.26_{\pm 0.05}$ | $0.31_{\pm 0.02}$ |
| R-BP | $0.76_{\pm 0.09}$ | $0.87_{\pm 0.03}$ | $1.05_{\pm 0.03}$ | $0.26_{\pm 0.01}$ | $0.37_{\pm 0.01}$ |
| $R_d$-BP | $0.40_{\pm 0.03}$ | $0.42_{\pm 0.00}$ | $1.00_{\pm 0.02}$ | $0.34_{\pm 0.02}$ | $0.27_{\pm 0.03}$ |
| AR-BP | $0.52_{\pm 0.13}$ | $0.42_{\pm 0.01}$ | $1.06_{\pm 0.02}$ | $0.21_{\pm 0.02}$ | $0.29_{\pm 0.02}$ |
| $AR_d$-BP | $0.37_{\pm 0.10}$ | $0.39_{\pm 0.01}$ | $0.99_{\pm 0.02}$ | $0.20_{\pm 0.02}$ | $0.28_{\pm 0.03}$ |
| ARnet-BP | $0.45_{\pm 0.11}$ | $\mathbf{-0.03}_{\pm \mathbf{0.00}}$ | $1.41_{\pm 0.07}$ | $0.24_{\pm 0.04}$ | $0.26_{\pm 0.04}$ |
| QB-Vine | $\mathbf{-0.81}_{\pm \mathbf{1.26}}$ | $0.54_{\pm 0.34}$ | $\mathbf{0.87}_{\pm \mathbf{0.20}}$ | $\mathbf{-1.85}_{\pm \mathbf{1.16}}$ | $\mathbf{-0.76}_{\pm \mathbf{0.28}}$ |

**Scalability of the QB-Vine:** In Appendix E.1, we assess the scalability of the QB-Vine, on large data sizes and dimensions, by fitting Gaussian mixture models with 20 random means and non-isotropic covariance in dimensions $d = 50$ to $d = 600$ with $n = 10000$ train and test sets. We compare our model to an RQ-NSF, reporting the LPS as well as the maximum mean discrepancy and the reverse Kullback–Leibler divergence to assess sample quality, showing superior performance.

# 6 Discussion

We introduced the Quasi-Bayesian Vine, a joint Bayesian predictive density estimator with an analytical form and easy to sample from. This extends the existing works on Quasi-Bayesian predictive densities, by using Sklar's theorem to decompose the predictive density into predictive marginals and a copula to model the high-dimensional dependency. This decomposition enables a two-part estimation procedure, employing Quasi-Bayesian recursive density estimation for the marginals and fitting a simplified vine copula for the dependence, resulting in a convergence rate independent of dimension for certain joint densities. We empirically demonstrate the advantage of QB-Vine on a range of datasets compared to other benchmark methods, showing excellent modeling capabilities in large dimensions with only a few training data.

However, there is potential for further improvements. The main bottleneck of the QB-Vine is the simplified vine, both computationally and methodologically. The non-uniqueness of the vine decomposition resulting in a search over an exponentially large model space during estimation and the hyperparameter selection of the KDE pair copulas could both lead to misspecified models. Further, our main assumption is the use of a simplified vine copula which is only an approximation to the true distribution. While these concerns stem from limited effective copula models for high dimensions being available, from a practical point of view, a simplified vine offers tractable and fast likelihood evaluations, and ultimately outperforms competitors as shown in experiments.

Future directions of this work include the incorporation of more effective copula models, or copulas accommodating different dependence structures [98, 73, 51]. Another exciting direction are developments of new recursive Quasi-Bayes methods that can be merged into a Quasi-Bayesian Vine model [33].

**Author Contributions:** David Huk wrote the code, ran the experiments, derived the proofs and wrote the paper. Yuanhe Zhang helped write the initial code and the appendix. Ritabrata Dutta conceptualised the project and David Huk formulated the concrete method. Ritabrata Dutta and Mark Steel jointly supervised the project and helped to write the paper.

**Acknowledgments:** We thank all the reviewers for their helpful feedback. We also thank Surya T. Tokdar, Fabrizio Leisen and Edwin Fong for useful discussions, further thanking Vaidehi Dixit for sharing Prticle Filter codes with us. David Huk is funded by the Center for Doctoral Training in Mathematical Sciences at Warwick. Ritabrata Dutta is funded by EPSRC (grant nos. EP/V025899/1 and EP/T017112/1) and NERC (grant no. NE/T00973X/1).

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

# A Copulas

Copulas are a widely adopted tool in statistics and machine learning for modelling densities, permitting the construction of joint densities in a two-step estimation process. One firstly estimates the marginal densities as if they were independent of each other, and secondly, models the *copula* which accounts for the dependence between dimensions.

**Definition 2.1.1 in [48] (Copula)**  *A copula distribution is a multivariate distribution function with standard uniform univariate marginal distributions, that is, $U(0, 1)$ margins.*

This approach is motivated through Sklar's theorem:

**Theorem A.1** (Sklar [91])**.**  *Let $\mathbf{P}$ be an $d$-dimensional distribution function with continuous marginal distributions $P^1, P^2, \ldots, P^d$. Then there exists a copula distribution $\mathbf{C}$ such that for all $\mathbf{x} = (x^1, x^2, \ldots, x^d) \in \mathbb{R}^d$:*

$$\mathbf{P}(x^1, \ldots, x^d) = \mathbf{C}(P^1(x^1), \ldots, P^d(x^d))$$

*And if a probability density function (pdf) is available:*

$$\mathbf{p}(x^1, \ldots, x^d) = p^1(x^1) \cdot \ldots \cdot p^d(x^d) \cdot \mathbf{c}(P^1(x^1), \ldots, P^d(x^d))$$

*where $p^1(x^1), \ldots, p^d(x^d)$ are the marginal pdfs, and $\mathbf{c}(P^1(x^1), \ldots, P^d(x^d))$ is the copula pdf.*

If the marginal distributions are absolutely continuous, the copula is unique. Consequently, one can decompose the estimation problem of learning $p(\mathbf{x})$ by first learning all the marginals $\{p^i\}_{i=1}^d$, and in a second step learning an appropriate copula model $c(u^1, \ldots, u^d)$, where $u^i := P^i(x^i), i \in \{1, \ldots d\}$ are the images of the $x^i$ under the cdf of each dimension. By applying cdf transformations marginally, the copula is agnostic to the differences between dimensions such as axis scaling, and purely focuses on capturing the dependence structure among them.

Most parametric copula models are only suited for two-dimensional dependence modelling and greatly suffer from the curse of dimensionality [27, 47]. The Gaussian copula is a popular parametric choice as it is well-studied and can be fitted quickly even to moderate dimensions [48]. However, it lacks the desired flexibility to capture more complex dependencies involving multiple dimensions. Among nonparametric copulas, Kernel Density Estimator (KDE) copulas [34] are commonly used. They apply an inverse Gaussian distribution to the observed $\{u^i\}_{i=1}^d$ to map them to a latent space and perform regular KDE on the latent density. However, this KDE copula method suffers from the poor scaling of KDE estimators in higher dimensions. Finally, deep learning copula models remain a nascent line of research and typically are more computationally expensive and sample-dependent due to their reliance on large models, with only a handful of candidate solutions such as [60, 56, 53, 3]. As such, current copula models are mostly limited to low to medium-dimensional modelling[54].

## A.1  Gaussian copula

A popular parametric copula model is the Gaussian copula. It assumes that the dependence between dimensions is identical to that of a Gaussian distribution with mean $\mathbf{0}$ and covariance matrix $\Sigma$:

$$c(u^1, \ldots, u^d) = \frac{\mathcal{N}_d(\Phi^{-1}(u^1), \ldots, \Phi^{-1}(u^d); \mathbf{0}, \Sigma)}{\prod_{i=1}^d \mathcal{N}(\Phi^{-1}(u^i); 0, 1)}.$$

As such, its only parameters are the off-diagonal entries of the covariance matrix $\Sigma$. In the case of $d = 2$, there is a single parameter to estimate for the Gaussian copula.

## A.2  Gaussian Kernel Density Estimator copulas

As the marginal distribution of a copula is uniform in $[0, 1]$, the support of the copula estimator is restricted to $[0, 1]^d$ and must satisfy the uniform marginal condition. It is fairly difficult to build such an estimator that fulfills both desiderata with high expressivity. Gaussian Kernel Density Estimation for copulas [13, 34] is a popular approach for such nonparametric copula models, which models the copula on a latent Gaussian space. We explain the approach in the following.

By the inverse sampling theorem, for any $\mathbb{R}$-valued continuous random variable $X$, applying the corresponding cumulative distribution function $F$ to $X$ results in $F(X)$ being uniformly distributed in $[0, 1]$. Thus, we can transform a uniform random variable into any continuous distribution using its inverse cumulative distribution function. In the copula estimation stage, samples from the copula already have uniform marginal distributions, meaning we can apply any inverse distribution $F^{-1}$ to each marginal sample value and obtain a corresponding latent marginal distribution. If $F^{-1}$ is the inverse standard normal distribution, then the latent distribution for each marginal will be normal and $\mathbb{R}$-valued with no uniformity restrictions.

Gaussian KDE for copulas applies inverse standard normal distributions to each marginal, resulting in a latent representation of the samples on a Gaussian space. As such, one can employ regular Gaussian KDE to estimate the copula density on this latent space. In the case of two-dimensional copulas, the ensuing estimator has the following expression:

$$\hat{c}(u, v) = \frac{\sum_{k=1}^{K} \phi\Big(\Phi^{-1}(u) - \Phi^{-1}(u_k); 0, b\Big) \cdot \phi\Big(\Phi^{-1}(v) - \Phi^{-1}(v_k); 0, b\Big)}{\phi\Big(\Phi^{-1}(u); 0, b\Big) \cdot \phi\Big(\Phi^{-1}(v); 0, b\Big)},$$

where $(u, v)$ and $(u_k, v_k)$ are both in $[0, 1]^2$, $\{(u_k, v_k)\}_{k=1}^{K}$ are observed copula samples, and $\Phi$ and $\phi$ are respectively the Gaussian distribution and density with mean $0$ and variance $b > 0$.

## A.3 Vine copulas

A vine copula is an efficient dependency modelling method which decomposes $d$-dimensional copula estimation into $d(d-1)/2$ bivariate copula estimation via structured conditioning [5]. Here we illustrate the decomposition by a 3-dimensional copula density $c_{U,V,W}$ for the random vector $(U, V, W)$:

$$c_{U,V,W}(u, v, w) = c_{U,V}(u, v) \cdot c_{V,W}(v, w) \cdot c_{U,W|V}\Big(C_{U|V}(u\,|\,v),\, C_{W|V}(w\,|\,v)\,\Big|\,v\Big),$$

where

- $C_{U,V}$ is the copula of $(U,\,V)$, $C_{V,W}$ is the copula of $(V,\,W)$;
- $C_{U,W|V=v}$ is the copula of $(C_{U|V}(U\,|\,v), C_{W|V}(W\,|\,v))$ conditional on $V = v$;
- $C_{U|V}$ is the conditional distribution of $C_{U,V}$ on $V$, and $C_{W|V}$ is the conditional distribution of $C_{V,W}$ on $V$.

Generally, the distribution of $C_{U,W|V=v}$ will change with different values $v$ and this will make the model relatively complex. Therefore, it is common to use the simplifying assumption by ignoring the conditioning of pair copulas and simply model them as unconditional bivariate densities:

$$c_{U,V,W}(u, v, w) = c_{U,V}(u, v) \cdot c_{V,W}(v, w) \cdot c_{U,W|V}\Big(C_{U|V}(u\,|\,v),\, C_{W|V}(w\,|\,v)\Big).$$

The rationale of the simplified assumption is studied in [42]. In this paper, we mainly focus on the regular vine copula (R-vine) with a simplified assumption. The construction of an R-vine copula has two basic ingredients: (1) a valid tree structure of all random variables, (2) the choice of family for bivariate copulas.

Before we introduce the tree structure which is valid to construct a R-vine copula, let us first rigorously define the random variable of a copula we want to estimate. Suppose $\mathbf{U} = (\mathbf{U_1}, \mathbf{U_2}, ..., \mathbf{U_d})$ is a $d$-dimensional random variable which is distributed as a copula $C$, then we have that each marginal random variable $U_i$ is uniformly distributed in $[0, 1]$-scale. For notational simplicity, we will use the index of a random variable as its notation instead. Denote $\mathcal{T} = \{\mathcal{T}_1, ..., \mathcal{T}_{d-1}\}$ as a sequence of $(d-1)$ trees in terms of $\mathcal{T}_i = (\mathcal{V}_i, \mathcal{E}_i)$. Here we use a set of two nodes to represent the corresponding edge in $\mathcal{T}_i$, i.e., $e = (a, b)$ if node $a$ and $b$ in $\mathcal{E}_i$ are linked. To construct a valid R-vine copula, $\mathcal{T}$ satisfies the following conditions:

- $\mathcal{T}_1$ is a tree with a set of edges $\mathcal{E}_1$ and a set of nodes $\mathcal{V}_1 = \{1, ..., d\}$;

- $\mathcal{T}_i$ is a tree with a set of edges $\mathcal{E}_i$ and a set of nodes $\mathcal{V}_i = \mathcal{V}(\mathcal{E}_{i-1})$ for $i = 2, 3, ..., d-1$, where $\mathcal{V}(E)$ denoted that the pair of nodes which are linked by an edge in $E$ is treated as a new node.

- If two nodes in $\mathcal{E}_{i+1}$ are linked by an edge in $\mathcal{V}_{i+1}$, then they must be linked to one common node in $\mathcal{T}_i$.

For $\forall\, e = (a, b) \in \mathcal{T}_i$ with $i \geq 2$, we define

$$c(e) = a \cap b, \qquad a(e) = a \setminus c(e), \qquad b(e) = b \setminus c(e).$$

Finally, we rigorously define the R-vine copula as follows.

**Definition A.2** (Regular Vine Copulas). A $d$-dimensional copula $C$ is a regular vine copula if there exists a tuple $(\mathcal{T}, \mathcal{C})$ such that

- $\mathcal{T}$ is a regular vine tree sequence with $(d-1)$ trees;

- $\mathcal{C} = \{C_e : e \in \mathcal{E}_i, i \in [d-1], C_e \text{ is a bivariate copula}\}$ is a family of bivariate copulas for each edge;

- For $\forall e \in \mathcal{E}_i$ and $\forall i \in [d-1]$, then $C_e$ is corresponding to the copula of $\left(a(e), b(e)\right)\bigg|\, c(e)$.

Therefore, the density function of R-vine $C$ can be expressed as

$$c_{(\mathcal{T},\mathcal{C})}(u_1, ..., u_d)$$
$$= \prod_{i=1}^{d-1} \prod_{e \in \mathcal{E}_i} c_{(a(e),b(e))|c(e)}(C_{a(e)|c(e)}(v_{a(e)}|v_{c(e)}), C_{b(e)|c(e)}(v_{b(e)}|v_{c(e)})).$$

Here we illustrate an R-vine copula density in five dimensions where we use different colors for the pair copulas corresponding to each of the $d - 1 = 4$ trees.

$$c_{\mathcal{T},\mathcal{C}}(u_1, u_2, u_3, u_4, u_5) = c(u_1, u_2) \cdot c(u_1, u_3) \cdot c(u_2, u_4) \cdot c(u_3, u_5)$$
$$\cdot c_{1,5|3}(u_{1|3}, u_{5|3}) \cdot c_{2,3|1}(u_{2|1}, u_{3|1}) \cdot c_{1,4|2}(u_{1|2}, u_{4|2})$$
$$\cdot c_{2,5|1,3}(u_{2|1,3}, u_{5|1,3}) \cdot c_{3,4|1,2}(u_{3|1,2}, u_{4|1,2})$$
$$\cdot c_{4,5|1,2,3}(u_{4|1,2,3}, u_{5|1,2,3}).$$

Specifying the tree structure for an R-vine decomposition is essential and plays an important role in pair-copula estimation through conditioning. An excellent overview is given in [17]. Notably, an R-vine decomposition is not unique for a given joint copula pdf. An appealing tree selection algorithm is proposed in [72], where authors derive a modified BIC criterion that prioritizes sparse trees while being consistent when the dimension $d$ grows at most at the rate of $\sqrt{n}$ where $n$ is the sample size.

# B  Martingale Posterior Distributions

Here we explain *martingale posterior* distributions as a justification of the Bayesian approach through a focus on prediction.

## B.1  The Bayesian Choice as a Consequence of Predictive Uncertainty

A common goal in statistics is the inference of a parameter or quantity $\theta$ by analysing data in the form of observations $(x_1, \ldots, x_n)$, $n \in \mathbb{N}$. The rationale for learning from data is that each observation provides information about the underlying process and parameter $\theta$, without which statistical analysis would be redundant. Indeed, consider a decision-maker basing their decision on their belief about $\theta$ (in an i.i.d. setting). Having observed data $(x_1, \ldots, x_n)$ and given the opportunity to observe an additional data point $x_{n+1}$, they would be assumed to accept, as they could refine their beliefs

on $\theta$. This process of updating one's beliefs based on data is at the core of the Bayesian approach. Equipped with an initial guess about the parameter of interest $\theta$ captured by the prior $\pi(\theta)$, the goal is the inference about the distribution of the parameter given observed data $(x_1, \ldots, x_n)$ which is encoded in the posterior density $\pi_{(n)}(\theta|(x_1, \ldots, x_n))$.

For the decision-maker to refuse, it would mean that the additional observation has no significant effect on their belief. This implies, as identified by [29] and [50], that there is a point where additional observations $(x_{n+1}, \ldots, x_N)$ provide no benefit to the knowledge update of $\theta$. Inspecting the Bayesian posterior in terms of observed data $x_{1:n} = (x_1, \ldots, x_n)$ and possible future observations to be made $x_{n+1:\infty}$, written as

$$\pi_{(n)}(\theta|x_{1:n}) = \int \pi_{(n)}(\theta, x_{n+1:\infty}|x_{1:n}) \ dx_{n+1:\infty},$$

one can expand the right-hand integrand by including the predictive density $p$ for future data, obtaining

$$\pi_{(n)}(\theta|x_{1:n}) = \int \underbrace{\pi_{(\infty)}(\theta|x_{1:\infty})}_{\text{Bayes estimate}} \cdot \underbrace{p(x_{n+1:\infty}|x_{1:n})}_{\text{predictive density}} \ dx_{n+1:\infty}.$$

Having rewritten the posterior density in this way, it becomes apparent that the uncertainty in the value of $\theta$, which is given by the Bayes estimate, is a consequence of the uncertainty surrounding the imputation of missing observations $x_{n+1:\infty}$ through the predictive. With this insight, unlike the traditional Bayesian construction of prior-likelihood, [29] proceed to replace the predictive density $p$ with a predictive mechanism using all available data to impute the missing $x_{n+1:\infty}$ (or at least impute $x_{n+1:N}$ for a sufficiently large $N$), and replace the Bayes estimate $\pi_{(\infty)}$ with an appropriate functional of the complete data $x_{1:\infty}$. This predictive mechanism used to impute the unobserved data $x_{n+1:\infty}$ given observed $x_{1:n}$ directly leads to the *martingale posterior* as the limiting distribution of functionals when the unobserved data has been imputed, see definition 1 in [29].

### B.2   The martingale posterior of the QB-Vine

In [29, 35], the authors are interested in predictive resampling (not considered in this paper for the QB-Vine), which means to progressively sample from $\mathbf{p}_{(n)}$ and using those samples instead of observations to continue the recursive construction of $\mathbf{p}_{(n+1)}$. For this predictive resampling to make sense (see conditions 1 and 2 in [29]), they show that it is sufficient for the sequence to be conditionally identically distributed. This condition is in turn shown to be equivalent to the following martingale condition in [9]:

**Theorem B.1.** *(Theorem 3.1 of [9]). A sequence of random variables $\mathbf{X}_1, \mathbf{X}_2, \ldots$ is c.i.d. if and only if its densities $\mathbf{p}_1, \mathbf{p}_2, \ldots$ are such that for $n \geq 0$ and all $\mathbf{x} \in \mathbb{R}^d$:*

$$\int \mathbf{p}_{(n)}(\mathbf{x}) \cdot \mathbf{p}_{(n-1)}(\mathbf{x}^n) d\mathbf{x}^n = \mathbf{p}_{(n-1)}(\mathbf{x}).$$

We show that our QB-Vine construction indeed satisfies this condition. The only assumption is the ratio of two consecutive vines must be 1, i.e., the dependence structure is constant between predictive steps. We note that these derivations hold for any marginal recursive construction of the form (2) and any copula density used for $\mathbf{x}_n$, but show it for the QB-Vine here. We interpret the condition of having the same dependence structure between steps as natural when data is supposed to come from the same data-generating process, which is indeed the circumstance in which we apply the QB-Vine. Further, given observations, the best guess of the multivariate copula is given by fitting it at the last iteration, which is our approach in practice. We write $\mathbf{c}^{(i)} = \mathbf{v}^{(i)}$ to avoid confusion with bivariate copulas and denote inputs to $\mathbf{v}^{(i)}$ as vectors $(P_1(x_1), \ldots, P_d(x_d)) = [P^i(x^i)]$. For clarity, we also write the mixture between the independence and Gaussian copula used in (2) as $\left[(1 - \alpha_n) + \alpha_n \cdot c_\rho(P_{(n-1)}(x), P_{(n-1)}(x_n))\right] = c^n(P_{(n-1)}(x^i), P_{(n-1)}(x_n^i))$. The proof is as follows:

*Proof.*

$$\int \mathbf{p}_{(n)}(\mathbf{x}) \cdot \mathbf{p}_{(n-1)}(\mathbf{x}^n) d\mathbf{x}^n$$

$$= \int \prod_{i=1}^{d} \{p_{(n-1)}(x^i) \cdot c^n(P_{(n-1)}(x^i), P_{(n-1)}(x_n^i))\} \cdot \mathbf{v}_{(n)}([P_{(n)}^i(x^i)]) \cdot \mathbf{p}_{(n-1)}(\mathbf{x}^n) \, d\mathbf{x}^n$$

$$= \prod_{i=1}^{d} \{p_{(n-1)}(x^i)\} \cdot \mathbf{v}_{(n-1)}([P_{(n-1)}^i(x^i)])$$

$$\cdot \int \prod_{i=1}^{d} \{p_{(n-1)}(x_n^i) \cdot c^n(P_{(n-1)}(x^i), P_{(n-1)}(x_n^i))\} \cdot \frac{\mathbf{v}_{(n)}([P_{(n)}^i(x^i)])}{\mathbf{v}_{(n-1)}([P_{(n-1)}^i(x^i)])} \cdot \mathbf{v}_{(n-1)}([P_{(n-1)}^i(x_n^i)]) \, d\mathbf{x}^n$$

$$= \mathbf{p}_{(n-1)}(\mathbf{x}) \frac{1}{\mathbf{v}_{(n-1)}([P_{(n-1)}^i(x^i)])} \int \prod_{i=1}^{d} \{c^n(P_{(n-1)}(x^i), u_n^i)\} \cdot \mathbf{v}_{(n)}([P_{(n)}^i(x^i)]) \cdot \mathbf{v}_{(n-1)}([u_n^i]) \, d\mathbf{u}^n$$

$$= \mathbf{p}_{(n-1)}(\mathbf{x}) \frac{1}{\mathbf{v}_{(n-1)}([P_{(n-1)}^i(x^i)])}$$

$$= \int \prod_{i=1}^{d} \{c^n(P_{(n-1)}(x^i), u_n^i)\} \cdot \mathbf{v}_{(n-1)}([u_n^i]) \cdot \mathbf{v}_{(n)}([(1-\alpha_n) \cdot P_{(n-1)}^i(x^i) + \alpha \cdot H(P_{(n)}^i(x^i)|u_n)]) \, d\mathbf{u}^n$$

$$= \mathbf{p}_{(n-1)}(\mathbf{x}) \frac{\mathbf{v}_{(n)}([P_{(n-1)}^i(x^i)])}{\mathbf{v}_{(n-1)}([P_{(n-1)}^i(x^i)])} = \mathbf{p}_{(n-1)}(\mathbf{x}).$$

The first equality is applying Sklar on $\mathbf{p}_{(n)}(\mathbf{x})$ and using recursion (2) on the ensuing marginal densities $p_{(n)}^i(x^i)$. The second equality is Sklar on $\mathbf{p}_{(n-1)}(\mathbf{x}_n)$. The third step is obtained by the substitution $du_n^i = p_{(n)}^i(x_n^i)dx$. Lastly, we use equation (3) for the cdf inside the copula. Then, the result follows by noticing that the bivariate copulas and the copula $\mathbf{v}_{(n-1)}([u_n^i])$ integrate to 1 by copula properties (see e.g. the proof of Theorem 6 in [9]), and $[(1 - \alpha_n) \cdot P_{(n-1)}^i(x^i) + \alpha \cdot H(P_{(n)}^i(x^i)|u_n)]$ integrates into $[P_{(n-1)}^i(x^i)]$ due to it being a martingale marginally for each $i$. $\quad\square$

Consequently, predictive resampling is also possible with our approach and is as simple as sampling from the fitted copula, and marginally updating each univariate R-BP. This can be done in parallel across dimensions, instead of sequentially as in [29, 35], which is computationally appealing and opens an interesting avenue for future work.

### B.3  Constructing a martingale posterior for the DPMM

As explained in Section 2, given a sequence of observations $(x_i)_{i\geq 1}$ from the data generating process, the Bayesian approach wants to obtain an infinite sequence of copula densities $(c_i)_{i\geq 1}$ to recursively update the Bayesian predictive distribution. However, finding such an infinite sequence of copulas isn't feasible in practice and the ensuing copula family is determined by the prior-likelihood pair one implicitly chooses. For example, from [43], we have the following copula sequences:

- If we select the likelihood and prior as $l(y; \theta) = \theta e^{-\theta y}$ and $\pi(\theta) = e^{-\theta}$, then

$$c_n(u, v_n) = \frac{(n+1)[(1-u)^{-1-1/n}(1-v_n)^{-1-1/n}]}{n[(1-u)^{-1/n} + (1-v_n)^{-1/n} - 1]^{n+2}},$$

which is a sequence of Clayton copula [16] with parameter $n^{-1}$.

- If we select the likelihood and prior as $l(y|\theta) = \phi(y; \theta, 1)$ and $\pi(\theta) = \phi(\theta; 0, \tau^{-1})$, then

$$c_n(u, v_n) = \frac{\phi_2(\Phi^{-1}(u), \Phi^{-1}(v_n); \mathbf{0}, \boldsymbol{\Sigma}_{\rho_n})}{\phi(\Phi^{-1}(u); 0, 1)\phi(\Phi^{-1}(v_n); 0, 1)},$$

where $\phi_2$ is the pdf of a bivariate normal distribution, $\phi$ is the pdf of normal distribution, and

$$\boldsymbol{\Sigma}_{\rho_n} = \begin{bmatrix} 1 & \rho_n \\ \rho_n & 1 \end{bmatrix}, \qquad \rho_n = (n + \tau)^{-1}.$$

Therefore, we obtain a sequence of Gaussian copulas with parameters $\{\rho_n\}_{n \geq 1}$. Further, if we additionally put a conjugate prior on the variance parameter $\sigma^2$ of the likelihood $l(y|\theta) = \phi(y; \theta, \sigma^2)$, then we will recover a sequence of Student-t copulas.

The issue of model mis-specification naturally arises here due to the selection of prior-likelihood pair. Fortunately, here we can employ the DPMM which has relatively high flexibility due to its nonparametric nature. For the DPMM, suppose the first observation $x_1$ arrives, then we can obtain the first predictive via the updating kernel

$$k_1(x, x_1) = \frac{\mathbb{E}\left[f_G(x) f_G(x_1)\right]}{\mathbb{E}[f_G(x)] \cdot \mathbb{E}[f_G(x_1)]},$$

where $p_0(x) = \mathbb{E}[f_G(x)]$. Then, we can derive the numerator as

$$\mathbb{E}\left[f_G(x) f_G(x_1)\right]$$

$$= \mathbb{E}\left[\sum_{j=1}^{\infty} \sum_{k=1}^{\infty} w_j w_k \, \phi\left(x; \theta_j, 1\right) \phi\left(x_1; \theta_k, 1\right)\right]$$

$$= \left(1 - \mathbb{E}\left[\sum_{i=1}^{\infty} w_i^2\right]\right) \mathbb{E}_{G_0}\left[\phi(x; \theta, 1)\right] \mathbb{E}_{G_0}\left[\phi(x_1; \theta, 1)\right]$$

$$+ \mathbb{E}\left[\sum_{i=1}^{\infty} w_i^2\right] \mathbb{E}_{G_0}\left[\phi(x; \theta, 1) \phi(x_1; \theta, 1)\right]$$

$$= \left(1 - \mathbb{E}\left[\sum_{i=1}^{\infty} w_i^2\right]\right) p_0(x) p_0(x_1) + \mathbb{E}\left[\sum_{i=1}^{\infty} w_i^2\right] \mathbb{E}_{G_0}\left[\phi(x; \theta, 1) \phi(x_1; \theta, 1)\right].$$

The first equality follows from the stick-breaking representation [89] of the DP, we can formulate $G$ as

$$G(\cdot) = \sum_{i=1}^{\infty} w_i \, \delta_{\theta_i}(\cdot)$$

where

$$w_i = v_i \prod_{j<i}(1 - v_j), \qquad v_i \overset{i.i.d.}{\sim} \text{Beta}(1, a), \qquad \theta_i \overset{i.i.d.}{\sim} G_0.$$

Then, the second equality follows from the condition that $\sum_{i=1}^{\infty} w_i = 1$ almost surely. Denote $\alpha = \mathbb{E}\left[\sum_{i=1}^{\infty} w_i^2\right]$, then we can write the updating kernel as

$$k_1(x, x_1) = (1 - \alpha) + \alpha \cdot \frac{\mathbb{E}_{G_0}\left[\phi(x; \theta, 1) \phi(x_1; \theta, 1)\right]}{p_0(x) p_0(x_1)}$$

$$= (1 - \alpha) + \alpha \cdot \frac{\phi_2(\Phi^{-1}(u), \Phi^{-1}(v_1); \mathbf{0}, \Sigma_\rho)}{\phi(\Phi^{-1}(u); 0, 1) \phi(\Phi^{-1}(v_1); 0, 1)}$$

$$= (1 - \alpha) + \alpha \cdot c_\rho(\Phi^{-1}(u), \Phi^{-1}(v_1)),$$

where

- covariance matrix $\Sigma_\rho = \begin{bmatrix} 1 & \rho \\ \rho & 1 \end{bmatrix}$,
- $\Phi^{-1}$ is the inverse cdf of standard normal distribution,
- $u = \mathbb{P}_0(x), v_1 = \mathbb{P}_0(x_1)$,
- $p_0(\cdot) = \phi(\cdot; 0, 1 + \tau^{-1})$,

- $c_\rho$ is a bivariate Gaussian copula density function with the correlation parameter $\rho = 1/(1+\tau)$.

Therefore, we can see that the copula $c_1(u, v_1)$ for the first updating step is a mixture of independent copula and Gaussian copula. However, we will lose the tractable form of the copula from the second updating step. According to [43], instead of deriving the explicit rule for the sequence of copula, we fix the correlation parameter $\rho$ and set $\alpha$ to be a $(0, 1)$-valued decreasing sequence $(\alpha_i)_{i \geq 1}$.

## C   Strictly Proper Scoring Rules

In probabilistic machine learning, a Scoring Rule (SR) measures the appropriateness of a distribution $\mathbb{P}$ in modelling an observation $\mathbf{x} \in \mathcal{X}$ through a score written as $S(\mathbb{P}, \mathbf{x})$. For (hyper) parameter estimation, suppose we aim to model the underlying distribution of an observation $\mathbf{x}$ using a family of distributions $\mathbb{P}_\theta$ parametrized by $\theta$, then we can use a SR $S$ to select the suitable $\theta$. If we assume that $\mathbf{x} \sim \mathbb{Q}$, then we can obtain the expected SR $\mathcal{S}$ via taking expectation w.r.t. $\mathbf{x}$ as

$$\mathcal{S}(\mathbb{P}, \mathbb{Q}) = \mathbb{E}_{\mathbf{x} \sim \mathbb{Q}}\Big[S(\mathbb{P}, \mathbf{x})\Big].$$

According to [38], we call $\mathcal{S}$ strictly proper if $\mathcal{S}(\mathbb{P}, \mathbb{Q})$ is minimized if and only if $\mathbb{P} = \mathbb{Q}$, i.e. for $\forall \mathbb{P} \in \mathcal{P}$ with $\mathbb{P} \neq \mathbb{Q}$ such that

$$\mathcal{S}(\mathbb{Q}, \mathbb{Q}) < \mathcal{S}(\mathbb{P}, \mathbb{Q}).$$

Considering our settings of marginal distributions, we have observed data $\mathbf{x}_{1:n} \overset{i.i.d.}{\sim} \mathbb{P}^*$, and we aim to model $\mathbb{P}^*$ using $\mathbb{P}_\theta$. More explicitly, we don't need to use $\mathbb{P}_\theta$ directly in the expected SR, instead we normally use its probability density function or samples from this distribution to evaluate. In general, we optimise

$$\begin{aligned}
\theta^* &= \underset{\theta \in \Theta}{\operatorname{argmin}} \mathbb{E}_{\theta \in \Theta} \mathcal{S}(\mathbb{P}_\theta, \mathbb{Q}) \\
&= \underset{\theta \in \Theta}{\operatorname{argmin}} \, \mathcal{S}(\mathbb{P}_\theta, \mathbb{Q}).
\end{aligned}$$

Since we do not have the complete population of $\mathbb{Q}$, we use the empirical SR $\hat{\mathcal{S}}$ instead, i.e.

$$\hat{\theta}^* = \underset{\theta \in \Theta}{\operatorname{argmin}} \mathbb{E}_{\theta \in \Theta} \hat{\mathcal{S}}(\mathbb{P}_\theta, \mathbb{Q}),$$

where $\hat{\mathcal{S}}(\mathbb{P}_\theta, \mathbb{Q}) = \frac{1}{n} \sum_{k=1}^{n} S(\mathbb{P}_\theta, \mathbf{x}_k)$. Under mild conditions, it can be proven that $\hat{\theta}^* \to \theta^*$ asymptotically in [19]. For any positive definite kernel $k(\,\cdot\,, \cdot\,)$, the Kernel Score [38] is given by

$$S_k(\mathbb{P}_{\theta, \mathbf{x}}) = \mathbb{E}[k(\mathbf{Y}, \mathbf{Y}')] - 2 \cdot \mathbb{E}[k(\mathbf{Y}, \mathbf{x})],$$

where $\mathbf{Y}, \mathbf{Y}' \sim \mathbb{P}_\theta$. If we set $k(\mathbf{x}, \mathbf{y}) = -||\mathbf{x} - \mathbf{y}||_2^\beta$, then we obtain the **energy score** which is strictly proper if $\mathbb{E}_{\mathbf{Y} \sim \mathbb{P}_\theta}||\mathbf{Y}||_2^\beta < \infty$. The energy Score is defined as

$$S_E^\beta(\mathbb{P}_\theta, \mathbf{x}) = 2 \cdot \mathbb{E}||\mathbf{Y} - \mathbf{x}||_2^\beta - \mathbb{E}||\mathbf{Y} - \mathbf{Y}'||_2^\beta, \quad \text{for } \beta \in (0, 2].$$

Here $\mathbf{Y}$ and $\mathbf{Y}'$ are i.i.d. samples from $\mathbb{P}_\theta$. Practically, given finite samples $\mathbf{y}_1(\theta), ..., \mathbf{y}_m(\theta) \overset{i.i.d.}{\sim} \mathbb{P}_\theta$ where $\mathbf{y}(\theta)$ denotes a sample from $\mathbb{P}_\theta$ which is a differentiable function of $\theta$, the unbiased estimate of the energy Score via Monte Carlo approximation is

$$\hat{S}_E^\beta(\mathbf{y}_{1:m}(\theta), \mathbf{x}) = \frac{2}{m} \sum_{j=1}^{m} ||\mathbf{y}_j(\theta) - \mathbf{x}||_2^\beta - \frac{1}{m(m-1)} \sum_{k \neq j} ||\mathbf{y}_j(\theta) - \mathbf{y}_k(\theta)||_2^\beta. \tag{8}$$

Similarly, we can derive the unbiased gradient of $S_E^\beta(\mathbf{y}_{1:m}(\theta), \mathbf{x})$ w.r.t. $\theta$ which is crucial for any gradient descent algorithm. For notational convenience, we define

$$g(\mathbf{Y}, \mathbf{Y}', \mathbf{x}) = 2 \cdot ||\mathbf{Y} - \mathbf{x}||_2^\beta - ||\mathbf{Y} - \mathbf{Y}'||_2^\beta,$$

then $S_E^\beta(\mathbb{P}_\theta, \mathbf{x}) = \mathbb{E}_{\mathbf{Y}, \mathbf{Y}' \sim \mathbb{P}_\theta}[g(\mathbf{Y}, \mathbf{Y}', \mathbf{x})]$. Next, we have that

$$\nabla_\theta S_E^\beta(\mathbb{P}_\theta, \mathbf{x})$$
$$= \nabla_\theta \mathbb{E}_{\mathbf{Y}, \mathbf{Y}' \sim \mathbb{P}_\theta}[g(\mathbf{Y}, \mathbf{Y}', \mathbf{x})]$$
$$= \mathbb{E}_{\mathbf{Y}, \mathbf{Y}' \sim \mathbb{P}_\theta}\left[\nabla_\theta g(\mathbf{Y}, \mathbf{Y}', \mathbf{x})\right]$$
$$\simeq \frac{1}{m(m-1)} \sum_{j=1}^m \sum_{k=1}^m \nabla_\theta g(\mathbf{y}_j(\theta), \mathbf{y}_k(\theta), \mathbf{x}) \cdot \delta_{\{j \neq k\}}$$
$$= \frac{2}{m} \sum_{j=1}^m \nabla_\theta \|\mathbf{y}_j(\theta) - \mathbf{x}\|_2^\beta - \frac{1}{m(m-1)} \sum_{k \neq j} \nabla_\theta \|\mathbf{y}_j(\theta) - \mathbf{y}_k(\theta)\|_2^\beta$$
$$= \widehat{\nabla}_\theta S_E^\beta(\mathbf{y}_{1:m}(\theta), \mathbf{x}).$$

Furthermore, if $\mathbb{P}_\theta$ and $\mathbb{Q}$ both are univariate distributions of $\mathbb{R}$-valued random variables, then the energy Score will be reduced to

$$S_E^\beta(\mathbb{P}_\theta, x) = 2 \cdot \mathbb{E}|Y - x|^\beta - \mathbb{E}|Y - Y'|^\beta, \quad \text{for } \beta \in (0, 2], \tag{9}$$

where $Y, Y' \sim \mathbb{P}_\theta$, where the expectations can similarly be approximated using samples from $\mathbb{P}_\theta$. Notice that this will become the Continuous Ranked Probability Score (CRPS) [96] when $\beta = 1$.

The energy Score is a strictly proper scoring rule [38, 19, 76] and is a special instance of the maximum mean discrepancy [39] as well as a statistical divergence. It has been used as an effective objective for copula estimation [53, 1, 51] and even for R-BP marginal predictives as shown in [52]. It has also enjoyed success as an objective for Normalising Flows [90] and generative models [78]. The energy Score is the only objective among Wasserstein $p$-metrics [57] that supports unbiased gradient evaluations [6, 76], has a known optimisation-free solution and features a faster convergence rate than similar integral probability metrics [94, 32].

## D Proofs

### D.1 Lemma 3.2

*Proof.* We follow the Definition 14.4-3 of stochastic boundedness from [12]. Begin by choosing $\delta \in (0, 1)$. From Proposition 1 in [29], we have for $\epsilon > 0$ and $M > n$, over the supremum of $x \in \mathcal{X}$:

$$\mathbb{P}\left(\left|P_{(M)}(x) - P_{(n)}(x)\right| \geq \epsilon\right) \leq 2 \exp\left(-\frac{\epsilon^2}{\frac{2\epsilon \alpha_{n+1}}{3} + \frac{1}{2}\sum_{i=n+1}^M \alpha_i^2}\right)$$

$$\Leftrightarrow \lim_{M \to \infty} \mathbb{P}\left(\left|P_{(M)}(x) - P_{(n)}(x)\right| \leq \epsilon\right) \geq \lim_{M \to \infty} 1 - 2 \exp\left(-\frac{\epsilon^2}{\frac{2\epsilon \alpha_{n+1}}{3} + \frac{1}{2}\sum_{i=n+1}^M \alpha_i^2}\right).$$

Next, we choose a value $\epsilon_n$ (where the subscript shows that this quantity is dependent on $n$) to have the appropriate probability on the right-hand side by enforcing:

$$\delta = 2 \exp\left(-\frac{\epsilon_n^2}{\frac{2\epsilon_n \alpha_{n+1}}{3} + \frac{1}{2}\sum_{i=n+1}^\infty \alpha_i^2}\right)$$

$$\Leftrightarrow 0 = \epsilon_n^2 + \log\left(\frac{\delta}{2}\right) \frac{2\alpha_{n+1}}{3} \epsilon_n + \frac{1}{2} \log\left(\frac{\delta}{2}\right) \sum_{i=n+1}^\infty \alpha_i^2$$

with solution

$$\epsilon_n = \frac{-\log\left(\frac{\delta}{2}\right) \frac{2\alpha_{n+1}}{3} + \sqrt{\left[\log\left(\frac{\delta}{2}\right) \frac{2\alpha_{n+1}}{3}\right]^2 - 2\log\left(\frac{\delta}{2}\right) \sum_{i=n+1}^\infty \alpha_i^2}}{2}.$$

Due to $\delta \in (0, 1)$ and $\alpha_i > 0 \,\forall i$, we have:

$$|\epsilon_n| = n^{-1/2} \frac{-\log\left(\frac{\delta}{2}\right) n^{1/2} \frac{2\alpha_{n+1}}{3} + \sqrt{\left[\log\left(\frac{\delta}{2}\right) n^{1/2} \frac{2\alpha_{n+1}}{3}\right]^2 - 2\log\left(\frac{\delta}{2}\right) n \sum_{i=n+1}^\infty \alpha_i^2}}{2}.$$

With the choice of $\alpha_i = (2 - \frac{1}{i})(\frac{1}{i+1})$, we see that $\lim_{n\to\infty} n^{-a}\alpha_i$ is bounded for all powers $a \geq -1$. As such, both $-\log\left(\frac{\delta}{2}\right) n^{1/2}\frac{2\alpha_{n+1}}{3}$ and $\left[\log\left(\frac{\delta}{2}\right) n^{1/2}\frac{2\alpha_{n+1}}{3}\right]^2$ will safely be bounded for large enough $n$. Similarly, due to the choice of $\alpha_i$, we have $\sum_{i=n+1}^{\infty}(\alpha_i)^2 = \mathcal{O}(n^{-1})$, meaning $n^{-a}\sum_{i=n+1}^{\infty}(\alpha_i)^2$ will be bounded for large enough $n$ as long as $a \geq -1$. Hence $-2\log\left(\frac{\delta}{2}\right) n \sum_{i=n+1}^{\infty} \alpha_i^2$ will also be bounded for large enough $n$.

Consequently, for our choice of $\delta$, there exists a finite $K > 0$ and a finite $N > 0$ such that:

$$\sup_{x\in\mathcal{X}} \mathbb{P}\left(\left|\frac{P_{(\infty)}(x) - P_{(n)}(x)}{n^{-1/2}}\right| \leq K\right) \geq 1 - \delta \quad \forall n > N.$$

$\square$

## D.2  Theorem 3.3

We prove the statement for general densities, which then naturally extends to predictive densities. The proof of this result is largely an adaptation of the simplified vine copula convergence result (Theorem 1 in [70]) but where no rate on marginal densities is required. As such, our proof shares an identical approach until the last part, where we deviate. We have a weaker result for the convergence of distributions and yet show in what follows that convergence of our copula estimator can be obtained even without convergence guarantees on marginal densities.

**Notation used in the proof**  We follow vine copula notation from Appendix A.3 and use superscripts with parenthesis to now differentiate between samples instead of predictive steps, following notational conventions of the literature. We define $h$-functions as the conditional distribution functions for pair copulas

$$h_{j_e|\ell_e;D'_e}(u \mid v) := \int_0^u c_{j_e,\ell_e;D'_e}(s,v)ds, \quad \text{for } (u,v) \in [0,1]^2.$$

Further, we refer to the true unobserved samples of pair-copulas as

$$U_{j_e|D_e}^{(i)} := F_{j_e \mid D_e}\left(X_{j_e}^{(i)} \mid \mathbf{X}_{D_e}^{(i)}\right), \quad U_{k_e|D_e}^{(i)} := F_{k_e|D_e}\left(X_{k_e}^{(i)} \mid \mathbf{X}_{D_e}^{(i)}\right), \tag{10}$$

for $i = 1, \ldots, n$. We also denote estimators and quantities obtained by application to these unobserved samples with a bar superscript, for example:

$$\bar{c}_{j_e,k_e;D_e}(u,v) := \bar{c}_{j_e,k_e;D_e}\left(u,v,U_{j_e|D_e}^{(1)},\ldots,U_{k_e|D_e}\right).$$

Finally, denote with a hat superscript all quantities and estimators obtained by using $\hat{U}_l^{(i)} := \hat{F}_l(X_l^{(i)})$ instead of the true unobserved samples used in Equation (10).

**Assumptions**  For completeness, we state assumptions about the marginal distribution estimator $\hat{P}$ as well as bivariate copula estimators $\hat{c}$, even though our practical choices from the main text respect these. We begin by stating the assumption about our marginal distribution estimator denoted $\hat{P}$:

- **A1:** The marginal distribution function estimator has the following convergence rate:

$$\sup_{x\in\mathcal{X}} \left|\hat{P}(x) - P(x)\right| = \mathcal{O}_p\left(n^{-1/2}\right).$$

Next, we state our assumptions about the pair copula estimator for completeness:

- **A2:** For all $e \in E_m, m = 1, \ldots, d-1$, with $-r$ the convergence rate of a bivariate KDE copula estimator, it holds:
  (a) for all $(u,v) \in (0,1)^2$,

$$\bar{c}_{j_e,k_e;D_e}(u,v) - c_{j_e,k_e;D_e}(u,v) = O_p\left(n^{-r}\right),$$

  (b) for every $\delta \in (0, 0.5]$,

$$\sup_{(u,v)\in[\delta,1-\delta]^2} \left|\bar{h}_{j_e|k_e;D_e}(u \mid v) - h_{j_e|k_e;D_e}(u \mid v)\right| = o_{\text{a.s.}}\left(n^{-r}\right),$$

$$\sup_{(u,v)\in[\delta,1-\delta]^2} \left|\bar{h}_{k_e|j_e;D_e}(u \mid v) - h_{k_e|j_e;D_e}(u \mid v)\right| = o_{\text{a.s.}}\left(n^{-r}\right).$$

- **A3:** For all $e \in E_m, m = 1, \ldots, d-1$, it holds:
  (a) for all $(u, v) \in (0, 1)^2$,

$$\widehat{c}_{j_e, k_e; D_e}(u, v) - \bar{c}_{j_e, k_e; D_e}(u, v) = O_p\left(a_{e,n}\right),$$

  (b) for every $\delta \in (0, 0.5]$,

$$\sup_{(u,v) \in [\delta, 1-\delta]^2} \left| \widehat{h}_{j_e | k_e; D_e}(u \mid v) - \bar{h}_{j_e | k_e; D_e}(u \mid v) \right| = O_{\text{a.s.}}\left(a_{e,n}\right),$$

$$\sup_{(u,v) \in [\delta, 1-\delta]^2} \left| \widehat{h}_{k_e | j_e; D_e}(u \mid v) - \bar{h}_{k_e | j_j; D_e}(u \mid v) \right| = O_{\text{a.s.}}\left(a_{e,n}\right),$$

  where

$$a_{e,n} := \sup_{i=1,\ldots,n} \left| \widehat{U}^{(i)}_{j_e | D_e} - U^{(i)}_{j_e | D_e} \right| + \left| \widehat{U}^{(i)}_{k_e | D_e} - U^{(i)}_{k_e | D_e} \right|.$$

- **A4:** For all $e \in E_m, m = 1, \ldots, d-1$, the pair copula densities $c_{j_e, k_e; D_e}$ are continuously differentiable on $(0, 1)^2$.

We note that A1 is satisfied by our marginal predictive estimator, as proved in D.1 while A2, A3, A4 are all satisfied by the KDE pair copula estimator, as shown in [70].

**Proof strategy**  We perform the proof in three parts. To obtain the final result, we first prove the convergence of pseudo observations to true observations through induction. We then rely on the aforementioned convergence to show that feasible pair-copula density estimators $\hat{c}_{j_e, k_e; D_e}$, and conditional distribution function estimators $\hat{F}_{j_e | D_e}$ and $\hat{F}_{k_e | D_e}$ are pointwise consistent. Lastly, these two results are combined to obtain the convergence of the joint copula estimator $\hat{c}$ to the true multivariate copula $c$.

**Part 1: Convergence of pseudo observations**  we start by proving a convergence rate of samples on the copula space obtained through marginal distributions to their true unobserved equivalent. That is, $\forall \, e \in E_1, \ldots, E_{d-1}, i = 1, \ldots, n$,

$$\widehat{U}^{(i)}_{j_e | D_e} - U^{(i)}_{j_e | D_e} = \mathcal{O}_p\left(n^{-r}\right), \quad \widehat{U}^{(i)}_{k_e | D_e} - U^{(i)}_{k_e | D_e} = \mathcal{O}_p\left(n^{-r}\right). \tag{11}$$

Starting with $e \in E_1$ (the conditioning set $D_e$ is empty), as a consequence of **A1** we obtain the bound

$$\left| \widehat{U}^{(i)}_{j_e} - U^{(i)}_{j_e} \right| = \left| \widehat{F}\left(X_{j_e}\right) - F\left(X_{j_e}\right) \right| \leq \sup_{x_{j_e} \in \Omega_{X_{j_e}}} \left| \widehat{F}\left(x_{j_e}\right) - \overline{F}\left(x_{j_e}\right) \right| = \mathcal{O}_p\left(n^{-r}\right).$$

To obtain the second part of (11) one can use identical arguments, providing the initial inductive hook. Next, assuming (11) holds for all $e \in E_m$ with $1 \leq m \leq d-2$, we extend the induction to $e \in E_{m+1}$. Recalling that pseudo-observations $e' \in E_{m+1}$ are equal to $\hat{U}^{(i)}_{j_e | D_e \cup k_e}$ or $\hat{U}^{(i)}_{k_e | D_e \cup j_e}$ for some $e \in E_m$, it follows by multiple triangle inequalities that

$$
\begin{aligned}
\left| \hat{U}^{(i)}_{j_e | D_e \cup k_e} - U^{(i)}_{j_e | D_e \cup k_e} \right| &= \left| \hat{h}_{j_e | k_e; D_e}\{\hat{U}^{(i)}_{j_e | D_e} | \hat{U}^{(i)}_{k_e | D_e}\} - h_{j_e | k_e; D_e}\{U^{(i)}_{j_e | D_e} | U^{(i)}_{k_e | D_e})\} \right| \\
&\leq \left| \hat{h}_{j_e | k_e; D_e}\{\hat{U}^{(i)}_{j_e | D_e} | \hat{U}^{(i)}_{k_e | D_e}\} - \overline{h}_{j_e | k_e; D_e}\{\hat{U}^{(i)}_{j_e | D_e} | \hat{U}^{(i)}_{k_e | D_e}\} \right| \\
&\quad + \left| \overline{h}_{j_e | k_e; D_e}\{\hat{U}^{(i)}_{j_e | D_e} | \hat{U}^{(i)}_{k_e | D_e}\} - h_{j_e | k_e; D_e}\{\hat{U}^{(i)}_{j_e | D_e} | \hat{U}^{(i)}_{k_e | D_e}\} \right| \\
&\quad + \left| h_{j_e | k_e; D_e}\{\hat{U}^{(i)}_{j_e | D_e} | \hat{U}^{(i)}_{k_e | D_e}\} - h_{j_e | k_e; D_e}\{U^{(i)}_{j_e | D_e} | U^{(i)}_{k_e | D_e}\} \right| \\
&= H_{1,n} + H_{2,n} + H_{3,n}
\end{aligned}
$$

Notice that for $\delta_i := \min\{U^{(i)}_{j_e | D_e}, U^{(i)}_{k_e | D_e}, 1 - U^{(i)}_{j_e | D_e}, 1 - U^{(i)}_{k_e | D_e}\} > 0$, we have that all realisations $(U^{(i)}_{j_e | D_e}, U^{(i)}_{k_e | D_e})$ are contained within $[\delta_i, 1 - \delta_i]^2$ almost surely. Similarly, all realisations

$(\hat{U}^{(i)}_{j_e|D_e}, \hat{U}^{(i)}_{k_e|D_e})$ are in $[\delta_i/2, 1 - \delta_i/2]^2$ for sufficiently large $n$ as a consequence of (11). Combining this with **A2** (b) and **A3** (b), with large enough $n$:

$$H_{1,n} \leq \sup_{(u,v)\in[\delta_i/2,1-\delta_i/2]^2} \left| \hat{h}_{j_e|k_e;D_e}(u|v) - \overline{h}_{j_e|k_e;D_e}(u|v) \right| = \mathcal{O}_p(a_{e,n}),$$

$$H_{2,n} \leq \sup_{(u,v)\in[\delta_i/2,1-\delta_i/2]^2} \left| \overline{h}_{j_e|k_e;D_e}(u|v) - h_{j_e|k_e;D_e}(u|v) \right| = \mathcal{O}_p(n^{-r}),$$

and by another application of (11),

$$a_{e,n} = \sup_{i=1,\ldots,n} \left| \hat{U}^{(i)}_{j_e|D_e} - U^{(i)}_{j_e|D_e} \right| + \left| \hat{U}^{(i)}_{k_e|D_e} - U^{(i)}_{k_e|D_e} \right| = \mathcal{O}_p(n^{-r}),$$

giving $H_{1,n} = \mathcal{O}_p(n^{-r})$. To complete part 1, we want to show that $H_{3,n} = \mathcal{O}_p(n^{-r})$. We write the gradient of $h_{j_e|k_e;D_e}$ as $\nabla h_{j_e|k_e;D_e}$ and use a first-order Taylor approximation of $h_{j_e|k_e;D_e}(\hat{U}^{(i)}_{j_e|D_e}|\hat{U}^{(i)}_{k_e|D_e})$ around $(U^{(i)}_{j_e|D_e}, U^{(i)}_{k_e|D_e})$ to get

$$H_{3,n} \leq \left| \nabla^\top h_{j_e|k_e;D_e}(U^{(i)}_{j_e|D_e}|U^{(i)}_{k_e|D_e}) \begin{pmatrix} \hat{U}^{(i)}_{j_e|D_e} - U^{(i)}_{j_e|D_e} \\ \hat{U}^{(i)}_{k_e|D_e} - U^{(i)}_{k_e|D_e} \end{pmatrix} \right| + o_{a.s.} \begin{pmatrix} \hat{U}^{(i)}_{j_e|D_e} - U^{(i)}_{j_e|D_e} \\ \hat{U}^{(i)}_{k_e|D_e} - U^{(i)}_{k_e|D_e} \end{pmatrix}$$

getting the desired result,. and hence the first equality of (11) by yet another application of (11). The second equation follows by identical steps, completing the induction.

**Part 2: consistency of conditional CDF and pair-copula density estimators** Following similar steps to as in Part 1, one can obtain that for all $e \in E_1, \ldots, E_{d-1}$, and all $x \in \Omega_X$, the CDF estimators are bounded as

$$\hat{F}_{j_e|D_e}(x_{j_e}|\boldsymbol{x}_{D_e}) - F_{j_e|D_e}(x_{j_e}|\boldsymbol{x}_{D_e}) = \mathcal{O}_p(n^{-r}),$$
$$\hat{F}_{k_e|D_e}(x_{k_e}|\boldsymbol{x}_{D_e}) - F_{k_e|D_e}(x_{k_e}|\boldsymbol{x}_{D_e}) = \mathcal{O}_p(n^{-r}). \tag{12}$$

To bound pair-copula density estimators, we apply the triangle inequality to obtain

$$\left| \hat{c}_{j_e,k_e;D_e}(u,v) - c_{j_e,k_e;D_e}(u,v) \right|$$
$$\leq \left| \hat{c}_{j_e,k_e;D_e}(u,v) - \overline{c}_{j_e,k_e;D_e}(u,v) \right| + \left| \overline{c}_{j_e,k_e;D_e}(u,v) - c_{j_e,k_e;D_e}(u,v) \right|$$
$$= R_{n,1} + R_{n,2}.$$

Assumption **A3** (a) coupled with (11) bounds $R_{n,1}$ while $R_{n,2}$ is bounded by**A2** (a), completing the second part.

**Part 3: Consistency of the vine copula estimator** Up to now, our steps have mirrored those of [70]. With the following we differentiate ourselves by noticing that to get a bound on the copula estimator alone, no marginal densities are required:

$$\hat{\mathbf{c}}(\boldsymbol{x}) = \prod_{k=1}^{d-1} \prod_{e \in E_k} \hat{c}_{j_e,k_e;D_e}\left\{ \hat{F}_{j_e|D_e}(x_{j_e}|\boldsymbol{x}_{D_e}), \hat{F}_{k_e|D_e}(x_{k_e}|\boldsymbol{x}_{D_e}) \right\}$$

$$= \prod_{k=1}^{d-1} \prod_{e \in E_k} \left[ c_{j_e,k_e;D_e}\left\{ \hat{F}_{j_e|D_e}(x_{j_e}|\boldsymbol{x}_{D_e}), \hat{F}_{k_e|D_e}(x_{k_e}|\boldsymbol{x}_{D_e}) \right\} + \mathcal{O}_p(n^{-r}) \right]$$

$$= \prod_{k=1}^{d-1} \prod_{e \in E_k} \left[ c_{j_e,k_e;D_e}\left\{ F_{j_e|D_e}(x_{j_e}|\boldsymbol{x}_{D_e}), F_{k_e|D_e}(x_{k_e}|\boldsymbol{x}_{D_e}) \right\} + \mathcal{O}_p(n^{-r}) + \mathcal{O}_p(n^{-r}) \right]$$

$$= \mathbf{c}(\boldsymbol{x}) + \mathcal{O}_p(n^{-r}). \qquad \Box$$

where the first line is a consequence of pair-copula estimator convergence in Part 2, and the second equality is a consequence of (12) and the fact that $c_{j_e,k_e;D_e}$ is continuously differentiable. This concludes the proof.

Table 3: Average LPS (in bpd, lower is better) over five runs with standard errors for the Digits dataset.

| Model
n/d | DIGITS
1797/64 |
|---|---|
| MAF | $-8.76_{\pm 0.10}$ |
| RQ-NSF | $-6.17_{\pm 0.13}$ |
| R-BP | $-8.80_{\pm 0.00}$ |
| $R_d$-BP | $-7.46_{\pm 0.12}$ |
| AR-BP | $-8.66_{\pm 0.03}$ |
| $AR_d$-BP | $-7.46_{\pm 0.18}$ |
| ARnet-BP | $-7.72_{\pm 0.28}$ |
| QB-Vine (30) | $-6.66_{\pm 0.16}$ |
| QB-Vine (50) | $-7.92_{\pm 0.23}$ |
| QB-Vine (100) | $-8.98_{\pm 0.34}$ |
| QB-Vine (200) | $-9.69_{\pm 0.48}$ |
| QB-Vine (300) | $-10.23_{\pm 0.10}$ |
| QB-Vine (400) | $-10.39_{\pm 0.13}$ |
| QB-Vine (500) | $-10.49_{\pm 0.20}$ |
| QB-Vine (full) | $-10.47_{\pm 0.28}$ |

## E   Experiments and practical details

Details of the UCI datasets are discussed in [35]. In experiments, We use the implementation of vine copulas from [88] through a Python interface. We follow the data pre-processing of [35] to make results comparable.

**Hyperparameter search**   In our experiments on small UCI datasets, we use a grid search over 50 values from 0.1 to 0.99 to select $\rho$, independently across dimensions, selecting possibly different values for each. To select the KDE pair copula bandwidth we use a 10-fold cross-validation to evaluate the energy score for 50 values between 2 and 4, as these ranges were appraised to give the best fits on preliminary runs on train data. We note the hyperparameter selection of $\rho$ also supports gradient-based optimisation (see Appendix C), which we utilise in the Gaussian Mixture Model experiments in Appendix E.1. In general, gradient-based optimisation of $\rho$ converges within less than five iterations. For energy score evaluations, with marginal predictives, we sample 100 observations and compare them to the training data, while for the copula we simulate 100 samples from the joint to compare with the energy score against training data.
For the PRticle filter, we took an initial sample size of $d \cdot n$ to accommodate for different dimensions while not being overcome by computational burden. The Kernel used is a standard multivariate Gaussian kernel.

**Compute**   We ran all experiments on an Intel(R) Core(TM) i7-9700 Processor. In total our experiments for the QB-Vine took a combined 15 hours with parallelisation across 8 cores, or 120 hours on a single core. The Digits dataset on 8 cores took us 6 hours to run with 5 different train and test splits. Other datasets require about half an hour for five runs in parallel, while the Gaussian Mixture study had a total time of 4 hours. The PRticle Filter takes about two hours on all density estimation tasks combined. The RQ-NSF experiments on Gaussian Mixture Models took about 4 hours combined. Our total compute time is therefore the equivalent of 126 hours on a single core. Our implementation of the QB-Vine is not fully efficient so the computational times are rough upper bounds.

**Selection of $P_{(0)}$ in practice**   In practice, the initial choice of $P_{(0)}$ is made to reflect the support and spread of the data. As we standardize our data to be mean 0 and have standard deviation 1, a natural choice is the standard Gaussian $\mathcal{N}(0, 1)$. However, given distribution transformations are used throughout the recursion, if observations fall in the tails of the predictive density, numerical overflow might make them redundant, lowering the accuracy of our approach. Therefore, it is desirable to have heavier tails than those of the true distribution to capture outliers accurately. This coincides with the theory on such recursion requiring heavier tails for the initial predictive compared to those of the data, see the assumptions on $P_{(0)}$ in [97, 64] and [43, 29]. As such, our default choice is a standard Cauchy distribution.

For experiments, we tested Normal, Cauchy, and Uniform (over the range of the training samples plus a margin) initial guesses on train data. Generally, the Cauchy distribution is a well-performing choice and obtained the best NLL in all but two experiments. We give a summary of initial density $p_0$ choices for different experiments in Table 4 for density estimation and Table 5 for regression and classification.

| Dataset | WINE | BREAST | PARKIN | IONO | BOSTON |
|---|---|---|---|---|---|
| Choice of $p_0$ | Cauchy | Cauchy | Cauchy | Normal | Cauchy |

Table 4: Choice of $p_0$ for different density estimation experiments.

| Dataset | BOSTON (reg) | CONCR (reg) | DIAB (reg) | IONO (class) | PARKIN (class) |
|---|---|---|---|---|---|
| Choice of $p_0$ | Normal | Cauchy | Cauchy | Cauchy | Cauchy |

Table 5: Choice of $p_0$ for different regression and classification experiments.

**Marginal Sampling**  Here we briefly introduce our inverse sampling method for marginal predictive distributions via linear interpolation. In general, through basic rules of probability, for univariate $x \sim p$ with the corresponding cumulative distribution function $P$ with inverse $P^{-1}$, given $u \sim \mathcal{U}[0, 1]$, we can obtain $\tilde{x} = P^{-1}(u)$ as a sample from $p$. However, in our case, we have an analytical expression for $P$ only, with no expression for $P^{-1}$. As such, to sample from our model, we need an approximation of $P^{-1}$ that we can evaluate.

To do so, we start by considering the range of values over which we seek to approximate $P^{-1}$. In our work, we consider a range $\mathcal{R} = [\min(x_{1:n}) - \eta, \max(x_{1:n}) + \eta]$ defined as the range from the lowest observation to the highest observation with an added extrapolation value $\eta$ to each side. This value can be adjusted depending on the extrapolation capabilities desired from model samples, and how heavy-tailed the data is thought to be. We found $\eta = 0.1$ to work well, given the data is pre-scaled to have standard deviation 1.

Next, for gridded, equally spaced ordered values $\{\bar{x}_i\}_{i=1}^{K}$ with $\bar{x}_1 = \min(x_{1:n}) - \eta$ and $\bar{x}_K = \max(x_{1:n}) + \eta$, we evaluate the cdf at each of these point. This gives us an equally sized set of points $\{\bar{u}_i\}_{i=1}^{K}$ with $\bar{u}_i := P(\bar{x}_i) \in [0, 1]$, where $K$ is a hyperparameter guaranteeing the exactness of our approximation with higher value of $K$ in exchange for an increased computational cost. To encompass the complete range of $[0, 1]$, we set $u_1 = 0$ and $u_K = 1$. We fix $K = 1000$ in our experiments.

We then use the set of gridded cdf values $\{\bar{u}\}_{i=1}^{K}$ to construct an approximation of $P^{-1}$ through linear interpolation. More specifically, given a value $u \in [\bar{u}_j, \bar{u}_{j+1}]$ with $1 \leq j \leq K - 1$ that we wish to evaluate the inverse at to obtain $\tilde{x} := P^{-1}(u)$, we have

$$\tilde{x} \approx \bar{x}_j + \frac{u - \bar{u}_j}{\bar{u}_{j+1} - \bar{u}_j}(\bar{x}_{j+1} - \bar{x}_j) := \widehat{P}^{-1}(u)$$

By consequence, we can easily obtain the gradient of $\tilde{x}$ w.r.t. marginal hyperparameter $\rho$ as

$$\frac{\partial \tilde{x}}{\partial \rho} = \frac{\partial}{\partial \rho}\left(\bar{x}_j + \frac{u - \bar{u}_j}{\bar{u}_{j+1} - \bar{u}_j}(\bar{x}_{j+1} - \bar{x}_j)\right) \tag{13}$$

$$= \frac{(\bar{x}_{j+1} - \bar{x}_j)}{(\bar{u}_{j+1} - \bar{u}_j)^2}\left(-(\bar{u}_{j+1} - u)\frac{\partial \bar{u}_j}{\partial \rho} - (u - \bar{u}_j)\frac{\partial \bar{u}_{j+1}}{\partial \rho}\right) \tag{14}$$

$$= \frac{(\bar{x}_{j+1} - \bar{x}_j)}{(P(\bar{x}_{j+1}) - P(\bar{x}_j))^2}\left(-(P(\bar{x}_{j+1}) - u)\frac{\partial P(\bar{x}_j)}{\partial \rho} - (u - P(\bar{x}_j))\frac{\partial P(\bar{x}_{j+1})}{\partial \rho}\right). \tag{15}$$

As such, gradients of the inverse cdf become gradients of the cdf, which we can efficiently compute with automatic differentiation software.

Table 6: Comparison of LPS for QB-Vine (our method) and RQ-NSF on GMM with 4 clusters for changing $n$ and $d$. Results for our QB-Vine method are shown as the top numbers of each row, and RQ-NSF values as the bottom numbers of each row.

| d \ n | 50 | 100 | 300 | 500 | $10^3$ |
|---|---|---|---|---|---|
| 10 | $\mathbf{3.98}_{\pm\mathbf{0.23}}$ $36.47_{\pm4.87}$ | $\mathbf{1.73}_{\pm\mathbf{0.29}}$ $17.14_{\pm1.51}$ | $\mathbf{2.15}_{\pm\mathbf{0.06}}$ $12.82_{\pm0.36}$ | $\mathbf{0.94}_{\pm\mathbf{0.31}}$ $7.10_{\pm0.26}$ | $\mathbf{2.43}_{\pm\mathbf{0.17}}$ $7.91_{\pm0.11}$ |
| 30 | - - | $\mathbf{17.94}_{\pm\mathbf{1.06}}$ $91.09_{\pm7.54}$ | $\mathbf{11.04}_{\pm\mathbf{0.35}}$ $50.51_{\pm2.20}$ | $\mathbf{12.87}_{\pm\mathbf{0.17}}$ $48.50_{\pm0.73}$ | $\mathbf{9.85}_{\pm\mathbf{0.40}}$ $34.98_{\pm0.31}$ |
| 50 | - - | - - | $\mathbf{38.59}_{\pm\mathbf{4.31}}$ $115.64_{\pm3.06}$ | $\mathbf{25.82}_{\pm\mathbf{0.06}}$ $112.16_{\pm2.05}$ | $\mathbf{26.14}_{\pm\mathbf{0.01}}$ $71.43_{\pm1.65}$ |
| 100 | - - | - - | - - | - - | $\mathbf{78.20}_{\pm\mathbf{0.23}}$ $268.88_{\pm1.37}$ |

## E.1 Comparison to normalising flow on Gaussian Mixture Model

We assess the performance of the QB-Vine on a mixture of 4 non-isotropic Gaussians across a range of dimensions and sample sizes. We simulate $n$ $d$-dimensional data points from

$$ p(\boldsymbol{y}) = \sum_{k=1}^{4} \pi_k \cdot \phi(\boldsymbol{y}; \boldsymbol{\mu}_k, \boldsymbol{\Sigma}_k) , $$

where $(\pi_1, \pi_2, \pi_3, \pi_4) = (0.2, 0.3, 0.1, 0.4)$ and

$$ \boldsymbol{\mu}_k \overset{i.i.d.}{\sim} \mathcal{U}[-50, 50]^d , \quad \boldsymbol{\Sigma}_k \overset{i.i.d.}{\sim} \text{Wishart}(d, \boldsymbol{I}_d) . $$

We compare the QB-Vine with the RQ-NSF as a benchmark off-the-shelf estimator. The hyperparameters for the RQ-NSF were chosen to give the best performance on training data, and are 100,000 epochs, 0.0001 learning rate, 1 flow step, 8 bins, 2 blocks, and 0.2 dropout probability in common. For the number of hidden features, we set 16 for $d = 10$, 32 for $d = 30$, 64 for $d = 50$, and 128 for $d = 100$. Our results in Table 6 show that the QB-Vine consistently outperforms the RQ-NSF for the dimensions and sample sizes considered.

We additionally considered some even higher dimensional examples to asses the QB-Vine's scalability. We study the performance in $d = 400, 500, 600$ dimensions on Gaussian mixture models (GMMs) with 20 random means (drawn uniformly from $[-100, 100]^d$) and non-isotropic covariances drawn from a Wishart distribution, with $n = 20000$ observations, and using a $50/50$ split for training and testing. We compare the QB-Vine against the RQ-NSF taken as a benchmark for high-dimensional modelling, with the same hyperparameters from the experiments of Table 6. We repeated this study 5 times with different seeds, leading to different GMM models. Figure 2 shows the LPS over the 5 runs for each method and dimension. Further, as the generating distribution is known, we sample from the fitted models and evaluate their samples under the true GMM density (known as the reverse KL divergence, lower is better), reported in Figure 3. Finally, we compute the Maximum Mean Discrepancy (MMD) as a commonly used measure that compares samples to observations, reported in Table 7. The MMD assess how close the model is to the true data-generating process by comparing model samples to observed data, with a lower MMD score implying a better fit for the data. The results suggest that the QB-Vine has better density estimation as well as sampling capabilities for these examples.

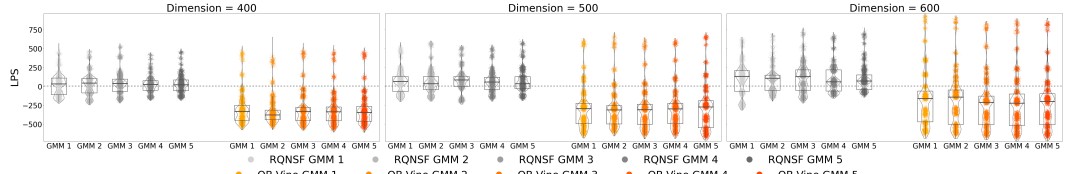

Figure 2: LPS (lower is better) for test points on 5 GMMs and for $d = 400, 500, 600$. The QB-Vine achieves lower LPS values on average than the RQNSF, across all 5 GMMs.

| Dimension | GMM 1 | GMM 2 | GMM 3 | GMM 4 | GMM 5 |
|-----------|-------|-------|-------|-------|-------|
| 400 | **29.2213** | **29.2775** | **29.0477** | **29.2993** | **29.3515** |
|     | 29.7847 | 29.7731 | 29.7247 | 29.7835 | 29.8447 |
| 500 | **32.7893** | **32.8354** | **32.5520** | **32.6044** | **32.7249** |
|     | 33.1789 | 33.4401 | 33.2011 | 33.4355 | 33.4143 |
| 600 | **35.7948** | **35.8328** | **35.9390** | **35.6756** | **35.7731** |
|     | 36.5586 | 36.6095 | 36.4700 | 36.4400 | 36.7090 |

Table 7: Comparison of the MMD (lower is better) computed on samples from the QBVine and RQNSF models across different dimensions and GMMs. Each cell shows the QBVine value on top and the RQNSF value on the bottom, separated by a dotted line. The QB-Vine outperforms the RQNSF in all cases considered, demonstrating better sample quality via this metric.

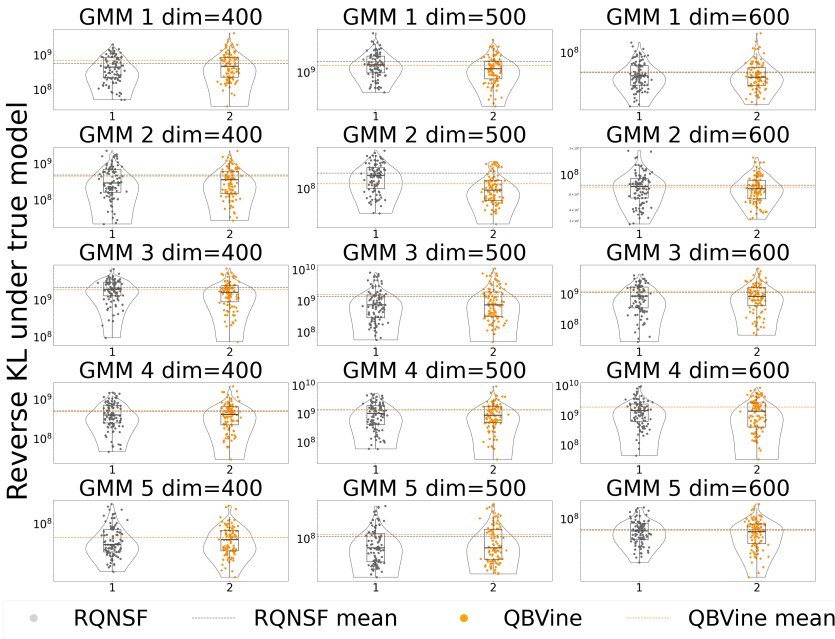

Figure 3: Reverse KL divergence (lower is better) for the GMM experiment in high dimensions, assessing sample fidelity. Both models perform equally well.

