# OpenReview forum: "Quasi-Bayes meets Vines"
_NeurIPS.cc/2024/Conference — NeurIPS 2024 poster_

### Official Review · Reviewer_qqFA · 2024-07-12

**Soundness:** 3
**Presentation:** 2
**Contribution:** 3
**Rating:** 5
**Confidence:** 4

**Summary:**

Quasi-Bayes, in the sense of building a model as a consistent sequence of predictive distributions, has recently received a lot of interest. The initial construction in [25] used products of bivariate copulas to build their predictives; the authors propose here to extend the construction to vine copulas, a flexible model for multivariate distributions.

**Strengths:**

* Quasi-Bayes in the wake of [25] is indeed a hot topic in Bayesian statistics, and the paper is potentially impactful.
* The initial copula constructions in [25] felt a bit constrained, and more flexibility would intuitively help.
* I overall enjoyed reading the paper, though I have many (hopefully constructive) comments below.

**Weaknesses:**

## Major
1. p1 L1 I would avoid flattering formulations like "[25] initiated a new era of Bayesian computation" and L33 "[25] heralded a revolution in Bayesian methods", "liberating Bayes". While you are free to think it and predict a revolution, that kind of statements about a 2023 paper are necessarily statements of opinion or personal predictions at best, and we should let time tell whether a new era or a revolution happened. You can say, e.g., that [25] proposed a stimulating change of paradigm, that is a fact. More pragmatically, starting a paper with opinion statements puts the reader in a suspicious mood.
2. Abstract: the sentence "But extensions [...] used" is unclear. What assumptions? What kernel?
3. Figure 1 is not particularly useful, I'd say, and the fonts in the decomposition are a bit strange. I know ICLR papers, for example, tend to have a small summary figure at the beginning, but I believe that it doesn't especially help here. The space gained could be used to give e.g. a more formal introduction to vine copulas in the main text.
4. p2 L42 "DPMM kernel structure". Maybe remind the reader on p1, when you first mention the DPMM, of what it is and what you call the kernel. It will not be obvious to many.
5. The introduction should contain a sentence defining vine copulas, or "vines". It is likely not a standard notion in the NeurIPS community, and the introduction should make it clear what the title of the paper means. Actually, the definition is somehow informally given in the caption of Figure 1, but I had missed it at first reading. I would avoid putting anything else in a caption than the description of what the figure shows. Important details should go to the main text.
6. p2 L56 "under the assumption of well-identified simplified vine copula model". It is unclear at this stage what you mean here.
7. Eqn 2: as far as I understand, going from the first to the second line requires Sklar's formula, at least if I follow the derivation of [25]. If you also use it, this should be mentioned. There is a reference to Appendix A, where copula tools are introduced, such as Sklar's theorem, but I'm not sure the argument is explained.
8. p3 L 82-82, in what sense does $c^{(n)}(\cdot,\cdot)$ converge to $1$? How does is guarantee a convergence for $p^{(n)}$? Why is the copula symmetric?
9. The paragraph L102--L113 is a bit dense and can gain in clarity. In what sense and under what assumptions does the "univariate R-BP model converge to a limiting distribution ... with density..."? What do you mean by "Newton's algorithm"? Also, there are steps to explain if you want to talk about how quasi-Bayes "approximates a posterior density"; so far, we have only talked about prediction. What parameter do you consider? How do you approximate the posterior density? You could refer to the corresponding passages in [25] and explain them in a few words. Or avoid discussing parameter inference if it is not central to your contribution.
10. p4 L128 what do you mean by "the predictives are unconditional marginal densities"? The sentence is puzzling.
11. Importantly, the authors of [25] restrict their designs of predictive distributions so that they satisfy their martingale condition (Eqn (4.1) in the Arxiv v2 version). They use their Corollary 1 for univariate copulas, and note in Section 4.3 that a multivariate extension of their Corollary 1 is not easy, which prompts them to take instpiration from  factorized kernels in DPMMs, for which they can guarantee (4.1). So the obvious question is: how do you guarantee that your vine predictives satisfy Fong el al.'s martingale condition? Or, alternatively, how do you guarantee the existence of the limiting martingale posterior $P^{(\infty)}$? It is possible that I am missing an obvious argument.
12. Lemma 3.2 uses $P^{(\infty)}$, which should be formally defined. Or at least, a reference should be given to the formal definition in [25]. But I think that giving a formal definition would help, since it would entail checking the martingale assumption of [25]; see bullet 11 above.
13. Overall, guarantees like Lemma 3.2 or Theorem 3.3 are guarantees in probability, under the distribution formed by the product of your predictives, am I right? In that case, these guarantees are meaningful only if I believe that the product of predictives is actually a good model for the data generating process. Is this realistic? Even under the classical simplifying assumption on conditional copulas that you are making?
14. p5 L146-173 the introduction to vine copulas is a bit confusing and informal. This section should be more formal, as understanding vine copulas is key to the paper, and I think not standard to many NeurIPS readers. I suggest having a subsection of Section 2 that formally introduces vine copulas, independently of their application to quasi-Bayes, and gives your example (23). Then Section 3 can focus on how you use vine copulas to define your joint predictive distributions. Also, I would use Pseudocode at the end of Section 3, to summarize all the steps in your procedure: hyperparameter tuning, estimation of the pair copulas, etc. I am not 100% confident that I can write this pseudocode from the text only.
15. Relatedly, having pseudocode would allow you commenting on the computational cost of the different steps. p9 L335 you mention "a search over a vast model space during estimation", but this has not been commented on before.
16. Theorem 3.3: what do you mean by "Assuming a correctly identified simplified vine strucgture for $c^{(\infty)}(\mathbf{u})$?". If I understand well, a more standard phrasing would be to formally introduce your model, say as $P$, and then state that under the distribution $P$, such and such is true.
17. The caption of Table 1 mentions error bars of two standard deviations averaged over 5 runs. I am unconfortable with the number of runs, which is intuitively too low for a $2\\hat{\sigma}$ interval to make sense, be it through a CLT or a Chebyshev argument.

## Minor
* p1 L29 the notation $\mathbf{p}^{n}(\mathbf{x})$ has not been defined yet, so I would simply avoid using the notation here.
* p2 L52 while you are not the only ones to do so, I don't see the need to boldface "main contributions".
* Theorem A.1: I would define what a copula distribution is.
* p3 L87 "intractable": following [25], I would even say it is not particularly desireable, as they try to bypass the need to specify a prior-likelihood pair.
* In Eqn (8), the densities should appear with their arguments.
* Section 5: don't you miss a log in your expressions of the log score?

**Questions:**

Q1. Can you give pseudocode for your method applied to a particular task, including all hyperparameter tuning steps like pair-copula estimation?

Q2. Can you then discuss the computational cost of each step, and identify the bottlenecks?

Q3. A convincing answer to items 11-12-13 in my list of major comments would also make me reconsider my mark.

**Limitations:**

This is fundamental work, and no immediate negative societal consequence is foreseen.

---

> ### Author Rebuttal · Authors · 2024-08-07
>
> # Reply 1:
> We did not mean to alienate readers, and will modify the text according to your suggestions. For the camera ready, we will use your phrasing for the abstract and use more neutral expressions in the introduction, citing papers emanating from [25] as evidence of an active area of research instead.
>
> Reply 2:
> Please see point 1 in the main reply.
> # Reply 3,5,6,14:
> We appreciate your judgement and removed the figure. We use the space to describe vines briefly in the introduction and thoroughly as a subsection of Section 2, accordingly correcting the relevant passages in the rest of the text.
>
> # Reply 4:
>  We thank the reviewer for this suggestion and have implemented it in the text in the style of point 1 of the main reply.
>
> # Reply 7:
>  We have added a sentence below the equation explaining the use of Sklar's theorem in the derivation by the definition of a copula.
>
> # Reply 8:
> In Corollary 1 of [25], for univariate predictives, it is shown that the existence of a copula sequence for predictive recursion is equivalent to the sequence of predictive densities satisfying the martingale condition, in turn implying that $p^{(n)}(x)\rightarrow p^{(\infty)}$ almost surely for each $x\in\mathbb{R}$ by the martingale convergence theorem. Due to the recursion of Equation (2), this implies $\lim_{n\rightarrow\infty}c^{(n)}(x,x^{n})=1\, a.s. \forall x$, meaning that the copula does not interfere with the convergence, and eventually the predictive does not change anymore. We have made the almost sure convergence explicit in writing and have justified it by saying "...$\lim_{n\rightarrow\infty}c^{(n)}(x,x^{n})=1\, a.s. \forall x$ as a consequence of the almost sure convergence of $ p^{(n)}$ with $n$ [25]". Lastly, the copula is symmetric as one can freely exchange the components of the copula in equation (2). Namely, exchange $f(x|\theta)$ with $f(x^n|\theta)$ for the expression of the joint density in the numerator and $p^{(n-1)}(x^n|x^{1:n-1})$ with $p^{(n-1)}(x|x^{1:n-1})$ for the marginals in the denominator, thus obtaining $c(x_1,x_2):=\frac{p(x_1,x_2)}{p_1(x_1)p_2(x_2)}=\frac{p(x_2,x_1)}{p_2(x_2)p_1(x_1)}=:c(x_2,x_1)$.
>
> # Reply 9:
>  We are quoting the result from theorem 5 in [25], saying that the R-BP distribution $P^{(n)}$ converges in total variation to $P^{\infty}$ which is absolutely continuous with respect to the Lebesgue measure with an associated density $p^{\infty}$. The theorem needs the R-BP density $p^{(n)}$ to be continuous, $\int_K {p^{(n)}}^2(x) d x<\infty$ for $K$ a compact subset of $\mathbb{R}$ with finite Lebesgue measure, the weight sequence to have the form
> $\alpha_i=\left(2-\frac{1}{i}\right) \frac{1}{i+1}$ and that the parameter of the copula be $\rho<1 / \sqrt{3}$. Then, in [39] the R-BP density is shown to converge to the true density in a Kulbach-Leibler sense, under similar assumptions. Newton's algorithm is the original work that initiated the study of recursive density estimates for the DPMM. The two cited papers [68,69] focus on establishing a recursion for the mixing distribution of a DPMM. However, at every step of the recursion their approach needs to solve an integral with dimensionality scaling with the dimension of the data, thereby making it impractical in higher dimensions. A thorough review of the original and ensuing works has been provided in [59]. We comment on this in Section 4. The "posterior density" was a typo, as we meant the predictive density, which is the object of the R-BP recursion. Parameter inference is indeed not central to our work, so we do not discuss it. We thank the reviewer for commenting on this and have ameliorated the clarity of the paragraph by referring to the corresponding results explicitly and fixing the typo about the posterior.
>
> # Reply 10:
> The goal was to emphasise that the univariate densities are not conditional as in [25,39], but are independent of each other allowing them to be modelled separately (i.e.~in parallel). We appreciate this comment on clarity and have rephrased the sentence into "the predictives are simple univariate densities".
>
> # Reply 11:
> In the interest of space, we include this as a comment below.
>
> # Reply 13:
> You are correct, our proofs are statements about the distribution formed by our model of the joint predictive, being marginal R-BPs multiplied with the vine copula. We do believe this is indeed a useful model with a rich complexity. Firstly, the marginal R-BP minimises the KL to the true data generating process with $n$ [39], and converges in total variation [25], leading to precise pseudo-observations $\{u_i=P^{(n)}(x_i)\}_{i=1}^d$ (lemma 3.2). Then, we employ copulas to recover the joint predictive, knowing that such a copula exists and is unique (theorem 3.1). We believe the best copula model currently available in the literature is a simplified vine copula, as it is both computationally feasible and offers a large model flexibility with non-parametric pair copulas (see e.g. \cite{czado2019analyzing}). Further, the convergence rates of the R-BP (lemma 3.2) and the vine copula (theorem 3.3) make them appealing density estimators.
>
> # Reply 14:
>  Please see the pseudo code in the pdf, and our point 2 in the main reply.
>
> # Reply 15:
>  We will include a discussion on the computational cost of our approach.
> -see reply to other reviewer and experiment on computing time-\\
> The model space we refer to is the specific decomposition of the vine copula, or equivalently its tree structure. We discuss this in L185-187. Following your comment, we will rephrase the sentence in Section 6 to be coherent with our discussion on computation costs mentioned above.
>
> Further replies and the proof are in the comment.

---

> ### Author Response · Authors · 2024-08-07
> **Rebuttal by Authors part 2 - proof**
>
> # Reply 11:
> We do not claim to converge to a multivariate $P^\infty(\mathbf{x})$. The $P^\infty(\mathbf{x})$ is only relevant when one wants to do predictive resampling to impute the unobserved data, as done in the martingale posterior for parameter inference. We do not investigate predictive resampling and do not advertise our work as such, nor focus on parameter inference. We need the marginals to converge (lemma 3.2). Then, for our theorem 3.3, this is purely a statement on the density as done in the vine copula literature, not on the limiting $P^\infty(\mathbf{x})$ martingale posterior in multiple dimensions. Similarly, theorem 3.1 adapts Sklar's Theorem to multivariate densities, holding for any $n$. For experiments, we show that our model approximates well all the datasets, doing density estimation, for which a $P^\infty(\mathbf{x})$ is not required. Our paper focuses on modelling predictive densities in an efficient way, which is a worthwhile pursuit as classical density estimators such as KDE are known to scale poorly with dimension.
> However, we were able to prove the martingale condition, see point 4 in the main reply, and the proof below. (We could not get it to format properly, we express our deepest apologies and hope it is still readable.) We write $\mathbf{c}^{(i)}= \mathbf{v}^{(i)}$ for clarity and denote inputs to copulas as vectors $(P_1(x_1),\ldots,P_d(x_d))=[P_i(x_i)]$:
> $$
> \begin{aligned}
> &\int \mathbf{p}^{(n)}(\mathbf{x})\cdot\mathbf{p}^{(n-1)}(\mathbf{x}^n)d\mathbf{x}^n\
> =&
>  \int \prod_{i=1}^d \left( p^{(n-1)}(x_i)\cdot c^{n}(P^{(n-1)}(x_i),P^{(n-1)}(x_i^n)) \right) \cdot \mathbf{v}^{(n)}([P_i^{(n)}(x_i)]) \cdot \mathbf{p}^{(n-1)}(\mathbf{x}^n) d\mathbf{x}^n\
> =&
> \prod_{i=1}^d\left(p^{(n-1)}(x_i)\right)\cdot\mathbf{v}^{(n-1)}([P_i^{(n-1)}(x_i)])\ \cdot \int\prod_{i=1}^d\left( p^{(n-1)}(x_i^n)\cdot c^{n}(P^{(n-1)}(x_i),P^{(n-1)}(x_i^n))\right)\cdot\frac{\mathbf{v}^{(n)}([P_i^{(n)}(x_i)])}{\mathbf{v}^{(n-1)}([P_i^{(n-1)}(x_i)])}\cdot\mathbf{v}^{(n-1)}([P_i^{(n-1)}(x_i^n)])d\mathbf{x}^n\=&\mathbf{p}^{(n-1)}(\mathbf{x})\frac{1}{\mathbf{v}^{(n-1)}([P_i^{(n-1)}(x_i)])}\int\prod_{i=1}^d\left(c^{n}(P^{(n-1)}(x_i),u_i^n)\right)\cdot\mathbf{v}^{(n)}([P_i^{(n)}(x_i)])\cdot\mathbf{v}^{(n-1)}([u_i^n]) d\mathbf{u}^n\=&\mathbf{p}^{(n-1)}(\mathbf{x})\frac{1}{\mathbf{v}^{(n-1)}([P_i^{(n-1)}(x_i)])}\int\prod_{i=1}^d\left(c^{n}(P^{(n-1)}(x_i),u_i^n)\right)\cdot\mathbf{v}^{(n-1)}([u_i^n])\cdot\mathbf{v}^{(n)}([(1-\alpha_n)\cdot P_i^{(n-1)}(x_i)+\alpha\cdot H(P_i^{(n)}(x_i)|u^n)])d\mathbf{u}^n\=&\mathbf{p}^{(n-1)}(\mathbf{x})\frac{\mathbf{v}^{(n)}([P_i^{(n-1)}(x_i)])}{\mathbf{v}^{(n-1)}([P_i^{(n-1)}(x_i)])}=\mathbf{p}^{(n-1)}(\mathbf{x}).
> \end{aligned}$$
> The first equality is applying Sklar on $ \mathbf{p}^{(n)}(\mathbf{x})$ and using recursion (4) on the ensuing marginal densities $p_i^{(n)}(x_i)$. The second equality is Sklar on $\mathbf{p}^{(n-1)}(\mathbf{x}^n)$ and writing out the recursive ratio of copulas for $\mathbf{v}^{(n)}$ (Equation (8) of the main text). The third step is obtained by the substitution $du_i^n=p_i^{(n)}(x_i^n)dx$. Lastly, we use equation (4) for the cdf inside the copula. Then, the result follows by noticing that the bivariate copulas and the copula $ \mathbf{v}^{(n-1)}([u_i^n])$ integrate to 1 by copula properties (see e.g. the proof of theorem 6 in [9]), and $[(1-\alpha_n)\cdot P_i^{(n-1)}(x_i) + \alpha \cdot H(P_i^{(n)}(x_i)|u^n)]$ integrates into $[P_i^{(n-1)}(x_i)]$ due to it being a martingale marginally for each $i$. Consequently, predictive resampling is also possible with our approach and is as simple as sampling from the fitted copula, and marginally updating each univariate R-BP. This can be done in parallel across dimensions, instead of sequentially as in [25,31], which is computationally much more appealing and opens an interesting avenue for future work. Thank you for your perceptive remark, which has guided us to new insights.

---

> ### Author Response · Authors · 2024-08-07
> **Rebuttal by Authors - part 3**
>
> # Reply 12:
> We note that $P^\infty(x)$ used in lemma 3.2 is univariate, with the martingale condition being established for the univariate R-BP in [25] already. Following your advice, in the camera ready, we will dedicate a paragraph to the introduction of the martingale condition and the proof of the QB-Vine satisfying it.
>
>
> # Reply 16:
>  We thank you for this comment, and have rewritten the statement with that structure.
>
> # Reply 17:
> Please see our reply to Q2 of reviewer iVqh, explaining our limitation due to [31]. With their code unavailable, we assume the best case scenario for other methods, believing those intervals are adequate there, but study the intervals of the QB-Vine in more detail, running 15 more runs for each point Figure 2 (see pdf).
> # Reply minor points:
>  We incorporated the changes in our manuscript.
>
> # Reply Q1:
> Please find the pseudo-code in Figure 2 of the pdf, including hyperparameter estimation. Some brief clarification: the training data can be permuted in full or partially depending on how much data is available, amd is captured by the variable $M$ giving the number of permutations. We keep $M=10$. Other hyperparameters are the number of points $B_1,\ldots,B_l$ to select the bandwidth of the vine. Our experiments were run with a grid of size $50$ between $2-4$ for UCI dataset and $0.5-3$ for the rest. $V=0.8$ to have $5-$fold cross-validation, and $J=100$. $\rho$ can be optimised with gradients and converged very quickly in our experiments, requiring less than five evaluations on average.
>
> # Reply Q2:
> The computational cost of each step comes from the existing method, R-BP or vines [65,], with the exception that we half the time of the R-BP [25,39] due to the optimisation of the Energy score. Due to space constraints, please see our reply to weakness 3 of reviewer iVqh.
>
>
> Finally, we thank the reviewer many times for their great suggestions and critical comments. We truly believe the paper is stronger as a result, and express our greatest thanks for dedicating your time to our work. We hope we addressed your concerns, and remain available to discuss any further points you would wish to raise.

---

> ### Comment · Reviewer_qqFA · 2024-08-10
>
> Thanks a lot for the detailed clarifications. Trusting the authors to include the proposed changes, I will increase my score to a 5. I am reluctant to increase more, because if I draw a parallel with a journal submission, for the latter I would have liked to review and carefully proofread a revised version of the paper.

---

### Official Review · Reviewer_2HTp · 2024-07-13

**Soundness:** 3
**Presentation:** 3
**Contribution:** 2
**Rating:** 6
**Confidence:** 4

**Summary:**

This paper develops the recently proposed quasi-Bayesian methods by applying vine copula (hence named QB-Vine) to the recursive Bayesian predicative distributions and bypassing the need for expensive posterior integration. The proposed method consists of two parts:  independent recursion of marginals by bivariate Gaussian copulas; and estimating the simplified vine copula to capture data dependence hence relaxing the kernel assumption for DPMM. Error bounds for both the distribution functions and the copulas are provided to justify the proposed QB-Vine. Numerical examples including density estimation and regression/classification are used to showcase the advantage compared with the state-of-the-art alternatives.

**Strengths:**

The paper proposes a novel Bayesian method to compute the predictive distribution. The proposed method, QB-Vine demonstrates numerical advantage over alternatives. Theoretic characterization on the errors is provided.

**Weaknesses:**

The dimension of the problems is relatively low (up to 64).

**Questions:**

Should there be $|\mathcal S_{ij}$ after the second conditional distribution in equation (10)?
Did you have dimension specific bandwidth $b$ and correlation $\rho$ for all the numerical experiments? Did it increase the overall computation time significantly by having them different for each dimension?
Line 223: "convergence" should be "converges"
Figure 2: why LPS increases from 500 samples to 900 samples for QB-Vine?

**Limitations:**

See weakness

---

> ### Author Rebuttal · Authors · 2024-08-07
>
> We thank the reviewer for their effort in reviewing and offer answers to their comments below.
>
> # Reply to weaknesses:
> To showcase the scalability of our approach in higher dimensions, we have expanded the experiments from Table 6 in the paper with a study in $d=400,500,600$ dimensions on Gaussian mixture models (GMMs) with 20 random means (drawn uniformly from $[-100,100]^d$) and non-isotropic covariances drawn from a Wishart distribution, with $n=20000$ observations, and using a $50/50$ split for training and testing. We compare the QB-Vine against the RQ-NSF taken to be a benchmark for high-dimensional modelling, with the same hyperparameters from the experiments of Table 6. We repeated this study 5 times with different seeds, leading to different GMM models. In the additional uploaded material, we included Figure 3 showing the LPS over the 5 runs for each method and dimension. Further, as the generating distribution is know, we sample from the fitted models and evaluate their samples under the true GMM density (known as the reverse KL divergence, lower is better), reported in Figure 1. Finally, we compute the MMD between samples and observations, reported in the table below.
>
> Dimension  GMM 1      GMM 2      GMM 3   GMM 4   GMM 5
> -------------------------------------------------------
> --------------------------------------------------------
> 400
> ------------------------------------------------------
> --------------------------------------------------------
> QB-Vine 29.2213    29.7847    29.2775 29.7731 29.0477
> ------------------------------------------------------
> --------------------------------------------------------
> RQNSF 29.7247    29.2993    29.7835 29.3515 29.8447
> -------------------------------------------------------
> --------------------------------------------------------
> 500
> ------------------------------------------------------
> --------------------------------------------------------
> QB-Vine        32.7893    33.1789    32.8354 33.4401 32.5520
> -------------------------------------------------------
> --------------------------------------------------------
> RQNSF           33.2011    32.6044    33.4355 32.7249 33.4143
> -------------------------------------------------------
> --------------------------------------------------------
> 600
> ------------------------------------------------------
> --------------------------------------------------------
> QB-Vine        35.7948    36.5586    35.8328 36.6095 35.9390
> -------------------------------------------------------
> --------------------------------------------------------
> RQNSF        36.4700    35.6756    36.4400 35.7731 36.7090
> -------------------------------------------------------
> --------------------------------------------------------
> \caption{Comparison of the MMD (lower is better) computed on samples from the QBVine and RQNSF models across different dimensions and GMMs. Each cell shows the QBVine value on top and the RQNSF value on the bottom, separated by a dotted line.}
>
> # Reply Q1:
> Yes, this is a consequence of the vine decomposition following Equation (9), meaning the copula density $c_{i,j}$ is itself conditional on $S_{ij}$. The vine copula of equation (10) is still a full vine copula, with no simplifying assumption. The simplifying assumption of a simplified vine copula then ignores that last conditioning in favour of a more parsimonious model. Additionally, the simplifying assumption makes the vine model easier to estimate, as then all the pairs $(P_{i|S_{ij}}(x_{i|S_{ij}}),P_{j|S_{ij}}(x_{j|S_{ij}}))$ can be used in the estimation of $c_{ij}$ instead of having to also take into account the values of the conditioning set $S_{ij}$. We take your comment as a potential confusion induced by our sentence on L154 "rewriting any conditional densities as copulas", and have therefore removed it for clarity.
>
> # Reply Q2
> We use a dimension-dependent correlation $\rho$ (L176), but use the same bandwidth for all KDE pair copulas (L181). We have modified L176 to state this more clearly.
>
>  # Reply Q3
>  The computational time is the same as estimating a common correlation parameter across all dimensions, since each recursion still has to be computed to select $\rho$. In fact, choosing a separate $\rho_i$ per dimension is preferred since it allows to estimate the parameters in parallel without each dimension needing to interact, for example to pool gradients as would be the case with a common bandwidth. Therefore, with parallelisation, selecting a different bandwidth for each dimension takes the same time as selecting the correlation for a single dimension.
>
> # Reply Q3
> We thank the reviewer for spotting this typo and have fixed it.
>
> # Reply Q4
> We believe this was a consequence of not taking enough runs in the experiment. We have now updated our picture to include 15 runs, showing a smooth decrease of the LPS with training size, as one would expect.

---

> > ### Comment · Reviewer_2HTp · 2024-08-08
> > **I have raised my score**
> >
> > Thanks to the authors for addressing my concerns. I appreciate the added results which make the numerics more convincing.

---

### Official Review · Reviewer_QJH9 · 2024-07-15

**Soundness:** 3
**Presentation:** 3
**Contribution:** 3
**Rating:** 7
**Confidence:** 2

**Summary:**

The authors propose a novel method for modeling high-dimensional distributions (for density estimation and supervised learning), where they break the estimation task into estimation of univariate marginals and estimation of a multivariate copula. To expedite the univariate estimation tasks they utilize the novel quasi-Bayesian (QB) estimation method, and they use a simplified vine copula to approximate the multivariate copula. They present empirical results that compare their method to others in density estimation and supervised learning.

**Strengths:**

- The authors present a well-defined problem that is of potential interest to conference readership, place it within the existing literature, and clearly demarcate their innovations.
- They break down the estimation problem into two sub tasks (estimation of the univariate marginals and the copula), and offer tangible contributions for both tasks, and provide convergence results for their estimator.
- Empirical results are convincing regarding authors' method's practical usefulness.

**Weaknesses:**

Although mostly outside of my expertise, this is a well-written paper overall. However, there are a number of potential changes that can improve the readability and accessibility of the paper. I list these in the section below.

**Questions:**

- L29: The paper makes a confusing start regarding its notation. $n$ is used here without being introduced, on L69 it's referred to as $K$, which leads to further confusion. This is likely the most important variable in the paper, so a proper introduction is required - especially because the recursive nature of the approach in question may not be as obvious for a reader from a different subfield.
- L163: Potential implications of using simplified vine copula can be discussed here, with a reference to forthcoming Thm 3.3.
- L174: Any particular reasons for the choice of Cauchy as the initial distribution?
- L188: Please use the additional space afforded after reviews to make this discussion more explicit.

**Limitations:**

The authors provide a satisfactory discussion of the limitations of the paper.

---

> ### Author Rebuttal · Authors · 2024-08-07
>
> We appreciate your effort in reviewing our paper and are glad you found it well-written - thank you. We provide answers to your suggestions below.
>
> # Reply Q1:
>  We thank the reviewer for pointing out this important oversight and have accordingly removed all $K$ and replaced them with $(n)$. We also removed the notation in the introduction and changed the indexing to be more standard, with dimensions as superscripts and recursive numbering as a subscript.
>
> # Reply Q2:
> We thank the reviewer for their comment. Following other reviewers' suggestions, for clarity's sake, we have moved the introduction of vine copulas to Section 2 as a subsection. We will incorporate your remarks there, detailing the implications of a simplified vine copula.
>
> # Reply Q3:
> The choice of initial predictive has 2 implications. Firstly, it is an implicit statement about your beliefs on observables, in a similar way to a prior. Secondly, it contributes to the efficiency of the recursion in fitting data. We discuss both of these aspects in Appendix E. In particular, a Cauchy distribution is effective at minimising numerical overflow when computing the cdf of data in the tails, making our algorithm more robust to different types of data. Further, taking a heavy tailed distribution such as the Cauchy coincides with the theoretical work on similar recursive density estimators, where it is common to assume that the tails of the true data generating density are not too heavy compared to the tails of the predictive, see e.g. condition (3.2) in [59].
>
> # Reply Q4:
> We thank you for this comment. We will expand upon this point in the camera ready.

---

> > ### Comment · Reviewer_QJH9 · 2024-08-12
> >
> > I thank the authors for their response. I believe the modifications they propose will improve the paper.

---

### Official Review · Reviewer_iVqh · 2024-07-19

**Soundness:** 2
**Presentation:** 3
**Contribution:** 2
**Rating:** 5
**Confidence:** 3

**Summary:**

Inspired by previous quasi-Bayesian (QB) methods, the recursive decomposition of Bayesian predictive posterior distributions, and vine copulas, the work introduces a new adaptation of QB to higher dimensions. The driving idea, introduced in Section 3 consists of making an adaptation of the previous decomposition for univariate densities to a higher number of dimensions, which ends up being a product of marginals times a vine copula with a certain design. Such vine copula models the conditioning among dimensions since the marginals do not. This one works via a decomposition into $d(d-1)/2$ elements that capture the dependency structure. Experiments on UCI datasets with a log-predictive metric show improvement wrt the selected SOTA methods with datasets of small N.

**Strengths:**

I would like to add some points of strength that I think are worth to be mentioned:

- The work introduces well enough the problem of higher dimension d in certain Bayesian inference problems and how some SOTA methods and frameworks might struggle when this is larger than a few dozens. Additionally, the reference to previous works and advances is important and well pointed out in my opinion, giving the right credit for the technical parts to each one of the previously published works.

- The paper is in general concise, and I could follow the technical details -- so I don't think there are details missing in the math part, despite some lack of analysis in certain directions that I'd had loved to see.

- In general, the idea of breaking the conditioning in this way, exploiting both recursive predictives and an additional element that captures correlations in a moderately-scalable way (i.e. vine copulas) is interesting to me and for the paper.

**Weaknesses:**

Some comments that I would like to add in terms of weak parts or ideas that I consider kind of a problem (or at least I'm concerned about):

- I think the technical derivations, and in general the notation could be improved. From the usual Bayesian perspective and for readers familiar with probabilistic methods, it is not really orthodox and the subscript-superscript system could be confusing at times. The work introduces too many things sometimes instead of focusing on clarity and highlighting the actual contribution and strong ideas proposed in the manuscript. At least, that's what I get from reading it.

- The contribution and novelty are a big concern to me. Honestly, I feel that if Eq. (7) is the big contribution or the point of novelty of the paper, it would not be enough, as it is just an adaptation of the previous derivations in Section 2 / Background to a vine copula. Additionally, the ratio in Eq. (8) is trivial, right? nothing really to derive there, or am I wrong here?

- The experiments are very limited, on small UCI datasets and only show one empirical result which is the improvement with the dimensions, however, no discussion about complexity/computational cost, run times, sensitivity to the choice of the vine copula system, etc is made. In that regard, I fear a bit how scalable really is the vine copula with such $d/(d-1)/2$ mechanism to capture conditioning.

**Questions:**

**Q1** -- What is the scalability of the simplified vine copula in L146-L152? Could the authors add some extra info about the computational cost per iteration n?

**Q2** -- Why so much focus on the log-predictive metric? Wasn't there an additional informative metric that could give the reader more insights about the performance compared with other inference methods? Is there a log() operator missing in the equation on L271?

**After rebuttal comment:** Dear authors, thanks for your rebuttal and the clarifications made regarding my points of weakness and concerns. Some things are clearer now to me, so I am happy to increase my score to 5 now. However, I still feel that some issues raised in my review have not or at difficult to fix atm, and I feel that there are not yet strong reasons to clearly accept the paper (i.e. clarity/readability and empirical results are yet limited, and I still have doubts on the novelty side). I don't object to acceptance, but won't be the supporter of that either. Best of luck.

**Limitations:**

Yes

---

> ### Author Rebuttal · Authors · 2024-08-07
>
> # Reply to weakness 1
> We appreciate your recommendation on this. Following your and other reviewer's comments, we have (1) interchanged subscripts and superscripts between dimension and predictive step, (2) added an introduction of vines as a subsection in Section 2, providing a clearer overview there and subsequently focusing more on our contribution on Section 3, along with minor fixes.
>
> # Reply to weakness 2
> The big contribution of our paper is the construction of a recursive QB method that is not restricted by assumptions on the DPMM kernel. As explained in point 1 of our main reply, the main drawback of previous methods is their necessary assumptions on the multivariate structure to obtain their multivariate recursion, leading to sub-optimal fits on multivariate data. Our tool to avoid restrictive assumptions on the multivariate structure is Sklar's theorem, which in itself is not new, but its application to bypass kernel assumption in a DPMM model is novel. Previous work [31] made incremental improvements to the recursive construction, whereas here we take a wholly different approach, overcoming the limitations of the existing literature in the field, and outperforming them. Additional points of novelty are the construction of a recursive estimator that can be computed in parallel instead of sequentially like the auto-regressive decomposition of [25,31], allowing multiple-fold computational savings, which is the primary focus of this line of research for the DPMM. This is further assisted  by optimising the energy score, halving our training time compared to existing work. Equation (8) is included there to make the potential for parallelisation and unrestricted dependence structure apparent.
>
> Our approach can further benefit from improvements to marginal recursion (as e.g. in [29] or future works since this is an active area of research as pointed out by reviewer qqFA) as well as improvement to copula models since both the R-BP and vines can be replaced with upgraded methods. This places our work in a unique intersection of recursive computations and multivariate copula models, connecting these up to now mostly disjoint subfields and opening a flow of cross-fertilisation between them.
>
> We thank you for your comment and take this as our queue to better motivate our contribution in the introduction. For the camera ready, we will add more context about the DPMM assumption (following point 1 in the main reply) in the text to clarify our innovation.
>
>
> # Reply to weakness 3:
> We appreciate your concern about scalability. We have included an example on Gaussian mixture models in high dimensions addressing this (see main reply). Further, to asses sensitivity, we expanded our study on the digits dataset, totalling 15 runs per each training size. We report that the change of copula structure does not negatively affect the final predictions, demonstrating robustness to the vine structure. We included a pseudo-code of our proposed algorithm in the pdf page.
>
> The computational cost of each step mirrors that of the existing methods, R-BP and vines [65], with the exception that we half the time of the R-BP [25,39] due to the optimisation of the Energy score. For the R-BP, our decomposition into marginals yields the same cost as the cheapest initial update step of [25] or [31] across all our recursive steps. Through parallelising, we obtain a constant cost with $d$, and scale as in [25,39], meaning $\mathcal{O}(n^2)$ for initialising the recursion (the first bullet point of the algorithm in the pdf), and $\mathcal{O}(n)$ to compute the pdf or cdf at a point. For the vine, the number of pair copulas grows quadratically with the dimension, and the number of tree structures is exponential with the dimension, leading to greedy algorithms being used for the tree selection, see [66]. To simulate from a vine, one can efficiently compute the inverse probability transformations (see e.g. Chapter 6 in [16]). Further, one can "truncate" the tree by assuming that pair copulas past a certain degree of conditioning are uniform, reducing the complexity significantly. One can also set a threshold of a simple association measure like Kendal's Tau, computable without model fitting, to decide which pair copulas to model and which to keep a priori. The drawbacks of the vine can be mitigated with truncation or thresholding as discussed above but inevitably loses out on modelling power. The flexibility of the vine in that regard has been studied in the literature, we refer to [16,17,65,66,67,87] among others. Vines are generally accepted as the best copula model for high dimensions, hence our use of it.
>
>
> # Reply Q1:
> Thank you for this important point. Following your question, we included an algorithmic description of our method (see reply to all reviewers) as well as a discussion on computational times and complexity.
>
> # Reply Q2:
> The LPS is a strictly proper scoring rule, meaning that the unique minimiser of the LPS is the true data generating mechanism. Further, the LPS is equivalent to the negative log-likelihood as commonly used to evaluate density estimators in the literature. Notably, the LPS is the only metric used in the previous QB literature [25,31]. As we compare with the results of [31], and their code has been removed from their website, we are unfortunately unable to replicate their experiments and evaluate other metrics. Following your insight, in our additional high-dimensional experiment on Gaussian Mixture models, we also compare the sampling quality of the model by computing the density of the samples under the model density (known as the reverse KL divergence) as well as the MMD, observing similar performance. There was indeed a missing log() on L271, we thank you for spotting it and have corrected it.

---

### Author Rebuttal · Authors · 2024-08-07

#### We thank all the reviewers for dedicating their time and effort to the review of our paper. Below, we outline the main points raised by reviewers and how we address them. We also uploaded a pdf file with extra results and a figure showing the algorithm for the QB-Vine. Further, we have posted an individual reply to each reviewer discussing their specific comments.

# Introduction to DPMM and main novelty of our approach:

Reviewers iVqh and qqFA have asked about the contribution of our work, and clarifications about statements related to previous work. We provide details here and have reformulated relevant parts of the abstract and introduction to clarify the novelty further.
Our paper addresses the recursive modelling of the DPMM's predictive density, also studied by previous work [25,31]. The multivariate DPMM is formulated as
$$f(\mathbf{x} \mid G)=\int_{\Theta} K(\mathbf{x} \mid \boldsymbol{\theta}) d G(\boldsymbol{\theta}), \text { with } G \sim \operatorname{DP}\left(c, G_0\right)$$
where $K$ is a kernel for the observables $\mathbf{x}\in\mathbb{R}^d$ parameterised by $\theta$, similarly to the kernel in kernel density estimation, and $G$ is called the mixing distribution, upon which a Dirichlet process prior is placed with base measure $G_0$ and concentration parameter $c>0$. The specification of $K$ and $G_0$ are thus important as they regulate the expressivity of the model, but their computation in a Bayesian model requires significant effort. To address this shortcoming, [25] provide a copula-based recursion obtained by assuming $K(\mathbf{x}|\theta)=\prod_{j=1}^d\mathcal{N}(x^j|\theta^j,1)$ and $G_0(\boldsymbol{\theta})=\prod_{j=1}^d\mathcal{N}(\theta^j|0,\tau^{-1})$, $\tau>0$, meaning both the kernel and base measure are assumed independent across dimensions, lacking the expressivity required to capture dependencies in the data. In [31], the form of the kernel is relaxed to be autoregressive with $K(\mathbf{x}|\theta)=\prod_{j=1}^d\mathcal{N}(x^j|\theta^j(\mathbf{x}^{1:j-1}),1)$ where the kernel mean $\theta^j:\mathbb{R}^{j-1}\mapsto\mathbb{R}$ is dependent on previous dimensions, and the base measure of [31] is a product of Gaussian Process priors $G_0(\boldsymbol{\theta})=\prod_{j=1}^d GP({\theta}^j|0,\tau^{-1}k)$ for $k:\mathbb{R}^{j-1}\times\mathbb{R}^{j-1}\mapsto\mathbb{R}$ a covariance function.

In our work, we posit the existing forms of $K$ and $G_0$, necessary for their recursions, lack the flexibility to model multivariate data accurately. As a result, we introduce the QB-Vine as a more general approach circumventing assumptions on the recursive form. Instead of deriving an autoregressive recursion through assumptions on $K$ and $G_0$, we show that by applying Sklar's theorem on the joint density, one only needs to specify the recursive form of the marginals, overcoming the limitations of existing work and leading to better performance in experiments. Lastly, our proposed method is inherently parallelizable over dimensions instead of sequential, leading to significant computational gains.

# Scalability to high dimensional problems:

 To showcase the scalability of our approach in higher dimensions, we have expanded the experiments from Table 6 in the paper with a study in $d=400,500,600$ dimensions on Gaussian mixture models (GMMs). We compare the QB-Vine against the RQ-NSF as a benchmark, repeating the experiment 5 times, and showing our competitive performance. In the uploaded pdf page, we show: the LPS (Figure 3), the reverse KL under the true model (Figure 1) and the MMD on samples (Table 1, see the reply to reviewer 2HTp). Further details are in the reply to weakness 1 of reviewer 2HTp.

# Computational Complexity:
We added pseudo code to the pdf for the QB-Vine. $M$ is the number of permutations, $B_1,...,B_l$ the bandwidth grid, $V$ the cross validation percentage and $J$ the number of samples to compute the Energy score in the vine. Please see our reply to Q1 of reviewer qqFA for more details. We further ran 10 additional runs on the digits dataset in Figure 4 of the pdf, to assess sensitivity. We discuss computational costs in detail in the reply to weakness 3 of reviewer iVqh.


# Checking the Martingale condition:
Reviewer qqFA asked to prove the martingale condition (4.1) in [25] for the QB-Vine, please see the proof in our reply 11 to them. The only assumption is the ratio of two consecutive vines must be 1, i.e., the dependence structure is constant between predictive steps. We note that these derivations hold for any marginal recursive construction of the form (2) and any copula density used for $\mathbf{c}^{(i)}$, but we focus here on the QB-Vine. We interpret the condition of having the same dependence structure between steps as natural when data is supposed to come from the same data-generating process, which is indeed the circumstance in which we apply the QB-Vine. Further, given $n$ observations, the best guess of the multivariate copula is given by fitting it at the last iteration, which is our approach in experiments.

---

### Author Response · Authors · 2024-08-13
**Closing comments**

We thank reviewers QJH9, 2HTp and qqFA for their positive review of our work and for increasing their scores. We believe this process has been fruitful in improving our manuscript and has led to a more cohesive paper. We kindly note the following improvements in response to the reviews:

For weakness 1 of reviewer iVqh, we have improved the clarity of our paper in-line with reviewers iVqh's comments and those in Q1/Q2 of QJH9 and in Major 3,5,6,14 of qqFA. Regarding contribution, reviewers iVqh's comment is echoed by Major 2 of qqFA. We thank reviewers for their favourable view of the contribution of our work, citing  "..., and clearly demarcate their innovations."-QJH9, "...a novel Bayesian method to compute the predictive distribution."-2HTp, "..., and the paper is potentially impactful."-qqFA. Lastly, we address comments on computational scalability, found in weakness 3/Q1 of iVqh, the comment on weakness 1 of 2HTp, and Major14/Q1/Q2 of reviewer qqFA.

---

### Decision · Program_Chairs · 2024-09-25

**Decision:**

Accept (poster)

**Comment:**

The authors tackle the problem of high-dimensional density estimation from modest amounts of data via a combination of two techniques: quasi-Bayesian inference, in which Bayesian updates are viewed in terms of copulas (pairing past data and a new observation) that are then approximated in a recursive process; and vine copulas (specifically, simplified vine copulas) in which multivariate distributions are assembled from bivariate copulas (Gaussian, in this paper).  The authors demonstrate superior performance compared to alternative methods on several problems with high dimension (in this case meaning 8-64 dimensional real data, 100-dimensional for one synthetic example in the appendices) and modest data relative to the dimension.  In the rebuttal period, the authors provided a comparison for an even higher dimension synthetic example (up to 600 dimensions).

Though surely more efficient than Monte Carlo-based alternatives, the inference procedure still involves an expensive search over a large model space (the space of possible nested trees is very large, and there are other hyper-parameters as well).  I note that while questions about scalability with dimension were surfaced during the discussion, there were few questions about scaling with data size -- though the authors point out that the one UCI test case where their method was beaten by a competitor involved a relatively large amount of data relative to the dimension.

The reviewers on this paper were weakly positive, though several expressed low confidence in their evaluation.  The most positive reviewers were the least confidence.  The other three landed on two weak accepts and one borderline.  Reviewers liked the combination of ideas, but were nervous about the utility for higher dimensions and data sizes, even with the updates in the rebuttals.  There was also some uncertainty about the novelty in the initial reviews, though this seems to have been largely resolved in the discussion period.

In general, machine learning audiences tend to favor methods that scale well to high dimensions and large data sets.  This paper focuses much more on what is large (in dimension and data size) with respect to the statistics literature, and this seems to be reflected in the reviewer reactions.  However, this is still an important regime, and this is an interesting combination of ideas, and the reviewers (and the AC, in my own reading) seem to agree with this.  For this reason, I am recommending acceptance.